# Active-passive microwave scattering in the Antarctica wind-glazed region: an analog for icy moons of Saturn

Léa Elise Bonnefoy[1,2], Catherine Prigent[1,7], Ghislain Picard[3], Clément Soriot[4,1], Alice Le Gall[5,6], Lise Kilic[7,1], and Carlos Jimenez[7,1]

[1]LERMA, Observatoire de Paris, PSL Research University, CNRS, Sorbonne Universités, Univ. Paris Cité, F-75014 Paris, France
[2]LMD/IPSL, Sorbonne Universités, ENS, PSL Research University, Ecole Polytechnique, Institut Polytechnique de Paris, Paris, France
[3]Institut des Géosciences de l'Environnement, Univ. Grenoble Alpes, CNRS, 38000 Grenoble, France
[4]University of Manitoba: Winnipeg, Manitoba, CA
[5]Laboratoire Atmosphères, Milieux, Observations Spatiales (LATMOS), UVSQ/CNRS/Paris VI, UMR 8190, 78280 Guyancourt, France
[6]Institut Universitaire de France, Paris, France
[7]Estellus, 93 boulevard de Sebastopol, 75002 Paris, France

**Correspondence:** Léa E. Bonnefoy (lb543@cornell.edu)

**Abstract.** Microwave radiometry and scatterometry, two complementary modes of sensing the composition and structure of the upper meters to hundreds of meters of the subsurface, are often difficult to reconcile, both on the Earth cryosphere and on icy moons of Saturn. To help interpret and model microwave scattering in porous, high-purity ices, we examine jointly 6.9 to 89 GHz AMSR2 radiometry in vertical (V) and horizontal (H) polarizations as well as 5.2 GHz ASCAT, 13.4 GHz QuikSCAT, and 13.5 GHz OSCAT scatterometry in the wind-glazed region of the East Antarctic ice sheet. The data are simulated using the Snow Microwave Radiative Transfer (SMRT) model, assuming a simplified snowpack characterized by constant temperature and a continuous increase in grain size (represented by optical radius) and density with depth. For the first time, we show that scatterometry and 6.9 to 37 GHz radiometry at V polarization can be successfully simulated with a unique simple snowpack model, indicating that incoherent volume scattering on subsurface heterogeneities dominates both the active and passive signals. To also simulate H-polarized radiometry, a thin surface ice layer as observed in the wind-glazed regions is one solution. Additional complexity, such as seasonal temperature variations, surface roughness, or non-continuous density variations, is necessary to explain the 89 GHz data and HH-polarized backscatter. Meanwhile, applying the same approach to simulate simultaneously passive and active Ku-band observations of icy moons improves on previous attempts but remains unable to reproduce the very high backscatter observed, highlighting the importance of coherent scattering and possibly unknown large icy structures (at least millimetric) in the subsurface. More work is still to be done to fully reproduce the microwave signatures of icy surfaces in the solar system.

# 1 Introduction

Planetary ices on Jupiter and Saturn's icy moons present an anomalous behavior to microwave radars. Indeed, their radar properties include very high backscatter – with Enceladus being the most radar-bright object in the solar system – and very high linear and circular polarization ratios (Hapke, 1990; Ostro and Shoemaker, 1990; Ostro et al., 2006, 2010; Black, 2001; Le Gall et al., 2019; Hofgartner and Hand, 2023). These properties are attributed to volume scattering in a low-loss medium, which is likely enhanced by coherent backscattering (Hofgartner and Hand, 2023, and references therein). However, Janssen et al. (2011); Bonnefoy et al. (2020); Le Gall et al. (2023) highlight another anomalous aspect of microwave scattering on Saturn's icy moons. So far radiative transfer calculations assuming purely-random scattering (e.g., Hapke, 1990, 2012; Janssen et al., 2016) have been unable to simultaneously reproduce active (radar) and passive (radiometric) observations at the same frequency. The Cassini radar instrument, operating in the Ku-band (13.78 GHz frequency; 2.2 cm wavelength), measured higher-than-expected backscatter pointing to the possible presence of organized efficient retro-scattering structures (Le Gall et al., 2023).

The surfaces and sub-surfaces of Saturn's icy moons, as observed by visible to microwave instruments, are constituted primarily of high-purity water ice, with a small quantity of unidentified non-icy material that varies regionally and from moon to moon (e.g., Le Gall et al., 2023, and references therein). Most observations indicate a composition of crystalline water ice, even if at the low surface temperatures (about 60 to 100 K) amorphous water ice would also be stable. At these temperatures, ice is unlikely to melt or metamorphize, although large impacts, micrometeorite gardening, and cryovolcanism can introduce heat and create large, though local and temporary, thermal gradients (Porter et al., 2010). Thermal infrared and microwave radiometry observations point to low thermal inertia and likely loose, porous ice created by impacts reworking the surface into a regolith (Howett et al., 2010; Ries and Janssen, 2015; Howett et al., 2016; Ferrari and Lucas, 2016; Bonnefoy et al., 2020). The very high radar backscatter from both Jupiter's and Saturn's moons is consistent with an icy, porous medium with multiple embedded scattering structures (Le Gall et al., 2019; Hofgartner and Hand, 2023). The high-porosity crystalline water ice of the surface and near-subsurface material (down to unknown depths varying from meters to hundreds of meters) is thus analogous to snow, even though its origin is very different: the impacts of any material onto icy moons (icy E ring dust, Phoebe ring dust, other impactors) occur at very high speeds, thus sand-blasting rather than "snowing on" the surface.

Icy moons have been explored using radar and radiometric observations, but without access to ground-truth data. Earth analogs of these remote environments can provide valuable in situ experiments and measurements to help interpret the remote sensing signatures obtained by radars and radiometers elsewhere in the solar system. To identify an appropriate analog site on Earth, we search for and study locations that resemble the surface conditions of icy moons as closely as possible. Such sites should offer both ground-truth information and comprehensive remote sensing datasets to support model configuration and hypothesis testing. Our primary selection criteria include regions that exhibit strong radar backscatter and are covered by dry, deep, and relatively stable snow (i.e., with minimal precipitation). Until now, Earth analogs used to interpret microwave radar observations of Jupiter's and Saturn's icy moons have included the Greenland percolation zone (Rignot et al., 1993; Rignot, 1995), penitentes in Chile (Hobley et al., 2018), and Northwest Greenland (Culberg et al., 2022). However, all of these analogs

present seasonal melt, a process which does not affect icy moons. Herein, we propose a new analog to understand microwave scattering on icy moon surfaces: the Antarctic wind-glazed and megadune regions. Their persistently low temperatures prevent melting of snow and ice (Traversa et al., 2023), while the high purity of the water ice, resulting from their distance from both the sea and rocky outcrops, closely resembles conditions on icy moons, particularly Enceladus (e.g., Le Gall et al., 2023).
Extremely low precipitation and near-zero snow accumulation (Traversa et al., 2023) approximate the almost atmosphere-less environments of moons, where little external material reaches the surface, allowing the formation of structures that remain stable for years. Moreover, the abundance of satellite observations combined with some in situ campaigns provides strong constraints for testing modeling approaches before applying them to the more uncertain conditions of icy moons.

More specifically, we focus on the large megadune field southward of Concordia station in the East Antarctic ice sheet (shown in Fig. 1), between 100 and 150°E, which is the most scattering and least emissive, non-melting region of Antarctica at 19 and 37 GHz (Fahnestock et al., 2000; Picard et al., 2009; Brucker et al., 2010). This region is the coldest and driest area on Earth (Traversa et al., 2023), with average temperatures around -50°C, and is swept by a constant katabatic wind which transports and sublimates snow downslope (Scambos et al., 2012). The snow megadunes are low-amplitude (2–4 m tall), long-wavelength (2–5 km spacing) eolian features with regions of snow accumulation separated by wide wind-glazed zones of near-zero net snow accumulation or even erosion (Frezzotti et al., 2002; Traversa et al., 2023). The glazed areas consist in a millimeter-thick ice crust covering a heavily metamorphized snow with large crystals (Albert et al., 2004; Courville et al., 2007). Due to the low accumulation rates, the snow within the top tens of centimeters to meters is exposed to seasonal temperature variations for decades or centuries, giving time for large hoar crystals to develop despite the low temperatures (Albert et al., 2004; Courville et al., 2007; Scambos et al., 2012). These large crystals are strong and efficient scatterers at microwave frequencies. They are responsible for the observed high radar backscatter (Fahnestock et al., 2000).

The wind-glazed regions of Antarctica have been associated with high radar backscatter, which is most likely due to large grain sizes in the subsurface (Fahnestock et al., 2000; Brucker et al., 2010) formed by the sublimation-deposition cycles over years (Courville et al., 2007). Because the deeper snow is older, it had time to produce larger depth hoar, forming a vertical gradient in grain size (Courville et al., 2007; Brucker et al., 2010). The depth probed by a microwave radiometer being approximately proportional to the wavelength, longer wavelengths (i.e., lower frequencies) probe deeper, where grains are larger in size in Antarctica. The longer wavelengths are thus subject to stronger scattering relatively to the homogeneous snowpack, leading to also to low emissivities. The shorter wavelengths are less affected by the vertical gradients. This leads to a flatter emissivity spectrum at 19 to 37 GHz as successfully modeled by Brucker et al. (2010) using the dense-medium radiative transfer model multi layer model (DMRT-ML) (Picard et al., 2013). Radiative transfer modeling has since been improved with the possibility to simulate active radar data simultaneously, leading to the Snow Microwave Radiative Transfer model (SMRT) thermal emission and radar backscatter model (Picard et al., 2018). This model has recently been applied to a multi-frequency dataset in several regions in Antarctica and the Canadian Arctic, where it is able to successfully reproduce brightness temperatures from 10 to 89 GHz using the subsurface structure measured on site (Picard et al., 2022a). Soriot et al. (2022) used the same model to simulate multi-frequency radiometry and C-band scatterometry over sea ice, highlighting the importance of depth hoar in multi-year sea ice. Herein, we apply this model to the Antarctic wind-glazed regions, for the first

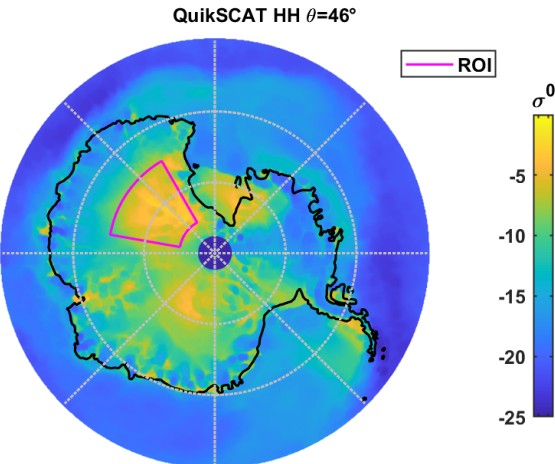

**QuikSCAT HH $\theta=46°$**

**Figure 1.** Yearly averaged QuikSCAT $46°$ incidence HH-polarization view of Antarctica. The region of interest (ROI), which is plotted in magenta, is the most radar-bright region of Antarctica.

time, to simulate both microwave thermal emission at five frequencies from 6 to 89 GHz and radar backscatter at 5.2 and 13.4 GHz.

The goals of this study are twofold : 1) to use a unique snowpack and radiative transfer model to simulate jointly horizontally (H) and vertically (V) polarized microwave radiometry from 6.9 to 89 GHz and scatterometry at 5.2 and 13.4 GHz in the Antarctica wind-glazed region, and 2) to infer novel interpretations on the comparative microwave scattering properties of cold icy surfaces, in Antarctica and on icy moons. In Section 2, we present the datasets used herein, which include observations from AMSR2 (Advanced Microwave Scanning Radiometer 2), ASCAT (Advanced SCATterometer), OSCAT (Oceansat SCATterometer), and QuikSCAT (Quick SCATterometer). Section 3 details the configuration of the SMRT model used to simulate these data, the snowpack properties, and the model parameters. Section 4 presents the success of the model in reproducing most observations, and the relative importance of each parameter in controlling the different frequencies, polarizations, or observation modes. Finally, Section 5 discusses the validity of the Antarctic megadunes region as an icy moon analog and applies the same method with a two-layer snowpack to attempt to reproduce simultaneously unresolved radar and radiometry observations of Saturn's satellites.

## 2 Remote sensing data

Most of the radar and microwave radiometry data on Saturn's icy moons were acquired with the Cassini Radar; it was thus crucial for the Earth analog dataset to include both active and passive data in the Ku-band. Additional frequencies, in both active and passive modes, help further constrain interpretations, clarify the spectral sensitivity to parameters such as grain size and density, and evaluate the model ability to successfully reproduce multi-frequency active and passive observations. Frequencies

| Instrument | ASCAT | AMSR2 | AMSR2 | QuikSCAT | OSCAT | AMSR2 | AMSR2 | AMSR2 |
|---|---|---|---|---|---|---|---|---|
| Mode | Active | Passive | Passive | Active | Active | Passive | Passive | Passive |
| Frequency (GHz) | 5.2 | 6.9 | 10.65 | 13.4 | 13.5 | 18.7 | 36.5 | 89.0 |
| Wavelength (cm) | 5.8 | 4.4 | 2.8 | 2.2 | 2.2 | 1.6 | 0.82 | 0.34 |
| Incidence angle | 25–65° | 55° | 55° | 46°(HH) | 49°(HH) | 55° | 55° | 55° |
|  |  |  |  | 54.1°(VV) | 57°(VV) |  |  |  |
| Polarization | VV | V&H | V&H | VV&HH | VV&HH | V&H | V&H | V&H |
| Mean resolution (km) | 50 | 48 | 31 | 12 | 13 | 18 | 9 | 4 |

**Table 1.** Main instrument characteristics.

above about 100 GHz probe very shallow depths (Picard et al., 2012) and are not sensitive to the subsurface properties.
Meanwhile, over very thick ice and snow, frequencies lower than 2 GHz (available for example with SMOS and SMAP) may require a coherent radiative transfer model to account for interference between layers (Leduc-Leballeur et al., 2015; Tan et al., 2015), whereas the SMRT models only incoherent scattering. We therefore restrict the analyzed data to frequencies from 5.2 to 89 GHz. The AMSR2, ASCAT, OSCAT and QuikSCAT datasets were all averaged over a year and gridded to a uniform resolution of 12.5 km/pixel. For each observation type, the swath data are projected over a 12.5 km grid using the EASE-grid
2.0 Southern hemisphere grid projection (Brodzik et al., 2012, 2014). All pixels falling within a given grid point are averaged over a full year of data, for each instrument and observation condition (frequency, polarization, incidence angle, and mode).

To identify dominant behaviors in the active and passive microwave datasets, we use the Kohonen classification algorithm (Kohonen, 1990), previously applied to a similar dataset in the Arctic by Soriot et al. (2022). This unsupervised machine learning algorithm integrates the multi-dimensional nature of the passive and active observations, and groups the observations into
115 clusters with similar features in the observation space. It synthetically describes the co-variability of the different observations. The clusters are self-organized with neighborhood requirements, meaning that when the classification has converged, nearby clusters have nearby characteristics in the observation space. The number of clusters is chosen so that at least one piece of observation (at one frequency, one polarization, and one mode) shows a clear statistical difference between clusters, i.e., the difference between clusters is greater than the standard deviation of that observation within these clusters. The classification in
10 clusters is shown in Figs. 2 and 3, whose comparison highlights the anticorrelation between active and passive data, with very radar-bright regions exhibiting low emissivities.

## 2.1 Multi-frequency radiometry: AMSR2

The passive radiometry data used herein were acquired by the Advanced Microwave Scanning Radiometer (AMSR2) aboard the Japanese polar orbiting satellite GCOM-W. This instrument, which has been operational since 2012, measures the brightness
temperature $T_B$ in V and H polarizations at frequencies from 6.9 to 89 GHz at an incidence angle of 55°. The resolution varies from 4 km at 89 GHz to 48 km at 6.9 GHz. Herein, we use the $T_B$ at frequencies of 6.9, 10.65, 18.7, 36.5, and 89 GHz (Maeda et al., 2016), which correspond to wavelengths of 4.4, 2.8, 1.6, 0.82, and 0.34 cm respectively.

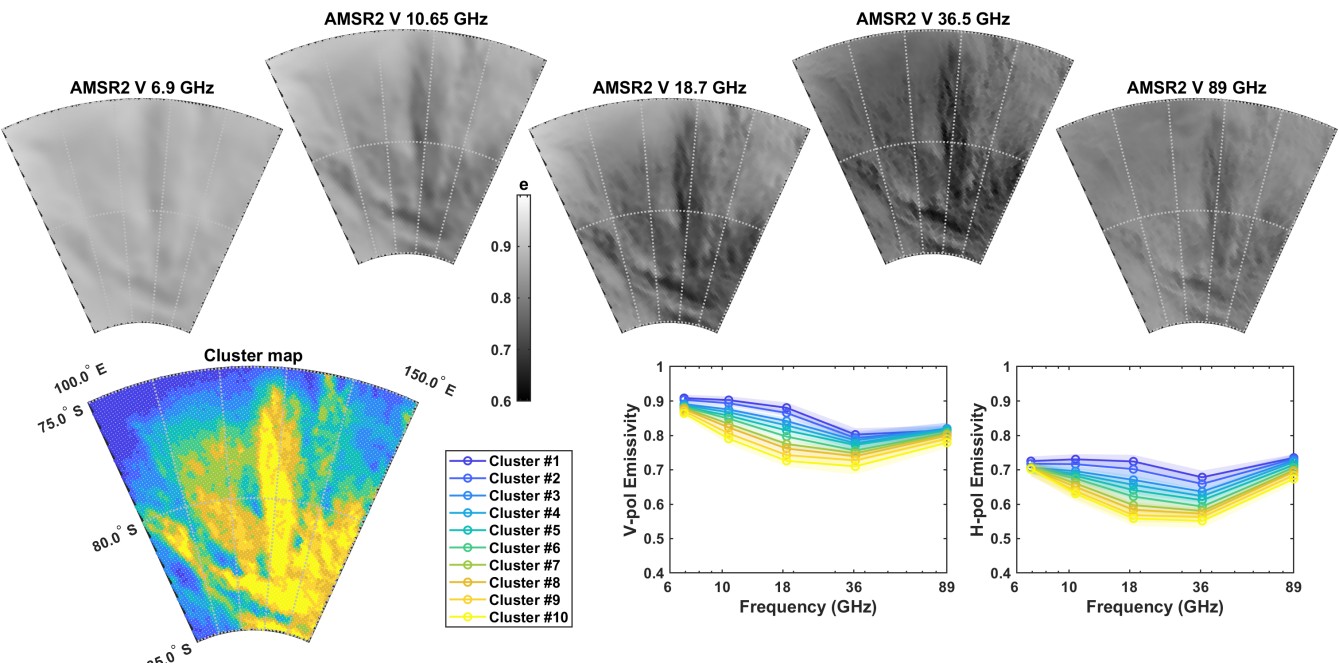

**Figure 2.** Top: AMSR2 emissivities at 6.9, 10.65, 18.7, 36.5, and 89 GHz in V polarization averaged over the year 2019 in the region of interest. Note the color scale is the same for all frequencies. Bottom: Result of the Kohonen clustering algorithm with 10 clusters. Note that the regions with lowest emissivity (cluster #10, in yellow) correspond to the radar-brightest regions and to the presence of megadunes.

AMSR2 emissivities have been computed using the AMSR2 brightness temperatures L1R data provided by the JAXA at their original spatial resolution for each frequency. The atmospheric contribution has been corrected using the radiative transfer model of Rosenkranz (2017). The inputs used to calculate the atmospheric contribution (atmospheric temperature and humidity profiles) have been provided by the ECMWF reanalysis data (ERA5). The surface temperature needed to compute the emissivity is the skin temperature provided by ERA5 data. To remove seasonal temperature variations, the emissivities (calculated for each AMSR2 observation each day with the collocated ERA5 atmospheric information and skin temperature) are averaged over the whole year of 2019. We only averaged emissivities calculated with low cloud liquid water content (less than $0.05 \ \mathrm{kg.m}^{-2}$, as indicated by the ERA5 reanalyses) to minimize cloud impact on the observed brightness temperatures. Pixels contaminated by Radio Frequency Interferences (RFI) are also filtered out using the flags provided by the JAXA. This dataset is shown in Fig. 2. Note that at a frequency where the atmosphere is transparent, the radiative transfer equation reduces to $T_B = e \times T$, and, for instance, a difference of 0.01 in emissivity $e$ with a snow / ice temperature of $T = 270$ K would result in a change of 2.7 K in $T_B$.

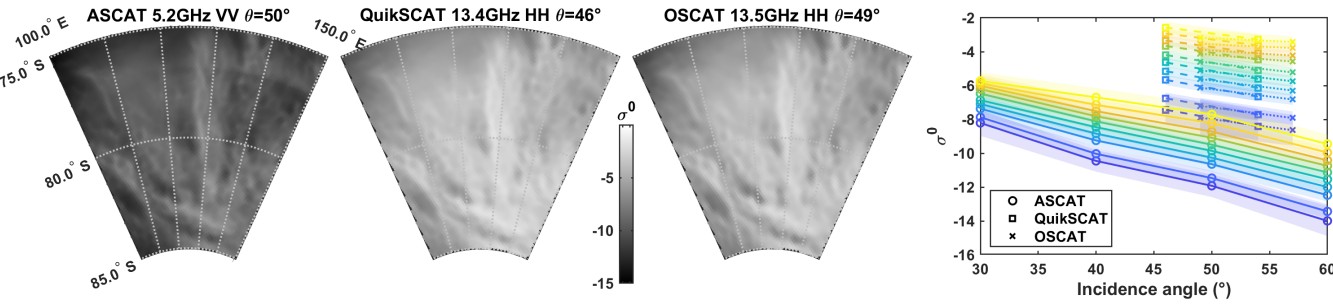

**Figure 3.** Visualization of the scatterometry $\sigma^0$: ASCAT (5.2 GHz), QuikSCAT (13.4 GHz), and OSCAT (13.5 GHz). The plot on the right shows the dependence with incidence angle for all three datasets, using the same clusters and colors as in Fig. 2. Note also that OSCAT and QuikSCAT are in HH polarization at the smaller incidence angles (46° and 49°, respectively) and VV at the larger incidence angles (54° and 57°), as detailed in Table 1.

## 2.2 C-band scatterometry: ASCAT

The Advanced SCATterometer ASCAT aboard the EUMETSAT's MetOp (Meteorological Operational) A, B, and C satellites operates in VV polarization at 5.2 GHz (C-band), at incidence angles from 25° to 65° and 50 km resolution (Figa-Saldaña et al., 2002). We used the publicly available level-1 calibrated normalized radar cross-section $\sigma^0$ measured by ASCAT. Radar observations are not directly sensitive to temperature and change very little with seasons in the region of interest (mean standard deviation=0.04 dB for ASCAT and 0.1 dB for OSCAT and QuikSCAT), but for consistency the ASCAT data were averaged over the year 2019 like the AMSR2 dataset. The ASCAT $\sigma^0$ data are shown in Fig. 3.

## 2.3 Ku-band scatterometry: QuikSCAT and OSCAT

The SeaWinds instrument aboard QuikSCAT measured the normalized backscatter cross-section $\sigma^0$ at 13.4 GHz (Ku-band) in HH polarization at an incidence angle of 46° and VV at 54°, at a resolution of $6 \times 25$ km, from 1999 to 2009. The Indian Space Research Organization's Oceansat SCATterometer (OSCAT) instrument aboard OceanSat-2 then ScatSat-1 also operates in Ku-band, at 13.515 GHz. $\sigma^0$ is measured in HH polarization at 49° and VV at 57°, at a resolution of $6 \times 30$ km. QuikSCAT and OSCAT data are both publicly available on the NASA Scatterometer Climate Record Pathfinder website (www.scp.byu.edu).

We use both datasets to improve coverage in terms of incidence angles; QuikSCAT data are averaged over 2008 and OSCAT over 2018. Although QuikSCAT did not observe over the same years as AMSR2, ASCAT, and OSCAT, the snow cover in the wind-glazed regions is unlikely to have substantially changed between 2008 and 2019; this is confirmed by the strong consistency between OSCAT and QuikSCAT (Lindell and Long, 2016; Hill and Long, 2017). The QuikSCAT and OSCAT $\sigma^0$ are shown in Fig. 3. In general, scattering is stronger at higher frequencies, for small scatterers such as depth hoar in snow and air bubbles in the ice. This explains why QuikSCAT and OSCAT data (Ku-band) exhibit higher scattering than ASCAT data (C-band).

## 3 Method

### 3.1 SMRT model configuration

We simulate the scatterometry and radiometry data using the SMRT model (Picard et al., 2018), which is capable of simulating both active and passive microwave data. The snowpack is modeled as a stack of plane-parallel horizontally infinite layers, each with fixed properties. The SMRT model allows for flexibility in the scattering model, the radiative transfer solver, and the microstructure model applied to each layer. Our specific choices and configuration are described below.

We use the symmetrized scaled strong contrast expansion theory (SymSCE) recently proposed by Picard et al. (2022b) based on theoretical work by Torquato and Kim (2021). This choice is explained by the fact that at 10 to 30 meter depths, which are probed at several frequencies considered herein (Table 2), the Antarctic snowpack densities lie around $450-550\,\mathrm{kg.m^{-3}}$, which is intermediate between snow and ice. However, most common scattering models, such as the improved Born approximation (IBA) or the dense-media radiative transfer quasi-crystalline approximation (DMRT-QCA), become inaccurate within this intermediate range, with a discontinuity between snow and ice (Picard et al., 2022b). The symmetrized version of the strong contrast expansion addresses this issue.

To solve the radiative transfer equation, we selected the Discrete Ordinate (DORT) method (Picard et al., 2018) which offers a consistent solution in both active and passive modes.

The snow microstructure model used for each layer is the scaled exponential as proposed in Picard et al. (2022b) with parameters defined in Section 3.2. The model outputs the H- and V-polarized brightness temperatures $T_B$ in K at each AMSR2 frequency, as well as the HH- and/or VV- polarized ASCAT, QuikSCAT, and OSCAT normalized radar cross-sections $\sigma^0$ at the frequencies and incidence angles of these instruments (see Table 1).

### 3.2 Snowpack model and parameters

Our snowpack model assumes monotonic variations in grain size and density with depth. Field measurements have shown that models with continuously increasing density and grain size fit reasonably well the observed behaviors (Albert et al., 2004; Courville et al., 2007; Brucker et al., 2010; Picard et al., 2014; Leduc-Leballeur et al., 2015; Inoue et al., 2024). These models do not account for the random and centimeter-scale vertical variations (e.g. Courville et al., 2007; Leduc-Leballeur et al., 2015; Picard et al., 2014; Inoue et al., 2024), which arise due to the specific conditions during each snowfall event or storm, and are especially pronounced in the upper few meters of the snowpack. Rather than identifying a realistic snow profile corresponding to field measurements, as previously done by e.g. Picard et al. (2022a), we aim to capture the general behaviors in grain size and density over large regions. We therefore assume that random and unresolved centimeter-scale fluctuations do not significantly affect the microwave observations presented in Section 2 (Leduc-Leballeur et al., 2015). The assumption of continuous variations in grain size and density is less valid in H polarization, which is sensitive to abrupt permittivity contrasts caused by density changes, and for higher frequencies, particularly 89 GHz which probes only centimetric depths (Picard et al., 2012).

| Frequency (GHz) | 5.2 | 6.9 | 10.65 | 13.4 | 13.5 | 18.7 | 36.5 | 89.0 |
|---|---|---|---|---|---|---|---|---|
| Maximum probed depths (m) | 22–88 | 22–102 | 9.4–39 | 2.8–11 | 2.7–11 | 2.6–11 | 0.4–3.4 | 0.1–0.7 |

**Table 2.** Estimated maximum depths probed at each frequency, calculated following Section 3.2. The range of values corresponds to the range of snowpack parameters given in Table 3. The depth is divided by 2 for frequencies corresponding to radar instruments (5.2, 13.4, and 13.5 GHz) to account for 2-way travel through the snowpack.

We simulate a 10-layer snowpack, with or without the presence of a thin, uniform ice crust on top. The maximum depth is chosen to be 200 m, at which the snowpack has always densified to solid ice (Hörhold et al., 2011; Leduc-Leballeur et al., 2015). The dielectric property of the ice follows Matzler et al. (2006, p. 456–461) (the default in SMRT) which is valid for the frequency range of 0.01-3000 GHz and ice temperatures from 20-273.15 K; the same formulation has been used on Jupiter's icy moons by Brown et al. (2023). Within the framework of the SMRT model with the exponential microstructure, each layer is characterized by four parameters: layer thickness, temperature, density, and optical radius.

**Layer thickness:** The SMRT model, which does not account for coherent effects, imposes that every layer must be thicker than $\lambda/4$, which for the lowest frequency (5.2 GHz; ASCAT) is 1.44 cm. Using the extreme low and high values of each parameter described above and in Table 3, we estimate the depths probed by each frequency. The maximum depth probed is calculated as the depth $z_i$ beyond which the optical depth of the above layers $\int_0^{z_i} \tau(z) dz$ is > 2; beyond this point, the structure of the snowpack does not significantly affect the signal at a given frequency. The optical depth of a given layer is calculated as $\tau = \kappa_e \times dz$, where $\kappa_e = \kappa_a + \kappa_s$ is the power extinction coefficient accounting for losses due to both absorption ($\kappa_a$) and scattering ($\kappa_s$), provided for each layer as an SMRT output, with $dz$ the layer thickness. The resulting maximum probed depths for the full range of parameters are provided in Table 2. Note that the minimum values of these depths correspond to large grain sizes leading to very high scattering ($\kappa_s$ dominates) whereas the highest values correspond to small grain sizes and very low scattering ($\kappa_a$ dominates). To fully sample the depths probed by each frequency, we chose 10 layers at exponentially increasing depths from 2 cm to 100 meters. Increasing the number of layers has no effect on the resulting emissivities and $\sigma^0$, indicating that the chosen sampling is sufficient. The bottom of the snowpack is modeled as a semi-infinite solid ice with density $917 \, \text{kg.m}^{-3}$.

**Layer temperature:** Because the microwave radiometry is averaged over a full year, we assume that the temperature is constant with depth, at least down to the depths probed at these frequencies. The temperatures are varied between -40 and -50°C. These values are typical of the region of interest, although they do not encompass the full range of annually averaged temperatures (-34° to -53°). Because we find that temperature influences both emissivities and backscatter values very little (Section 4), we do not test more values herein.

**Layer density:** The density variation with depth follows a simple exponential model reaching the value for solid water ice at depth (Bingham and Drinkwater, 2000; Leduc-Leballeur et al., 2015):

$$\rho(z) = \rho_{ice} - (\rho_{ice} - \rho_{top})e^{-Bz} \tag{1}$$

where $\rho_{ice} = 917$ kg.m$^{-3}$ is the density of water ice, $\rho_{top}$ is the value at the surface, and $B$ is an empirical value in m$^{-1}$, whose value depends on the snow type (Bingham and Drinkwater, 2000). Leduc-Leballeur et al. (2015) find $B = 0.017$ m$^{-1}$ at Dome C in Antarctica. This density profile matches well those found in Antarctic firn regions with low accumulation, where the pore close-off occurs deeper than is typical elsewhere (Van Den Broeke, 2008; Hörhold et al., 2011; Leduc-Leballeur et al., 2015).

**Layer optical radius:** Previous work has found that, for the grain sizes used in microwave radiometry simulations to be comparable to the optical radius measured in the field, a corrective factor must be used (Brucker et al., 2011; Royer et al., 2017). Recently, Picard et al. (2022b) has identified this factor as the polydispersity $K$, an intrinsic property of the snow microstructure, which can be measured from e.g. micro-computed tomography (Coléou et al., 2001). More precisely, we use the microwave grain size, defined by Picard et al. (2022b) as:

$$l_{MW} = K \times l_P \tag{2}$$

where $l_P$ is the Porod length, calculated from the layer density $\rho$, ice density $\rho_{ice} = 917$ kg.m$^{-3}$ and optical grain radius $r_{opt}$ as follows (Picard et al., 2022a):

$$l_P = \frac{4}{3}(1 - \rho/\rho_{ice})r_{opt} \tag{3}$$

where $r_{opt}$ is the radius of spheres having the same surface area over volume ratio as the considered snow microstructure (Grenfell and Warren, 1999). It is considered as a measurable quantity (Painter et al., 2007; Gallet et al., 2009; Picard et al., 2022a).

By comparing AMSR2 observations to SMRT simulations in a region where the vertical structure of the snowpack had been measured, Picard et al. (2022b) found that the best value for $K$ for Antarctic snow is 0.62. We therefore use this value herein, making the optical grain size $r_{opt}$ directly comparable to field measurements but this does not affect our results as fitting $l_P$ or $l_{MW}$ given a constant $K$ is strictly equivalent.

We model the increase of optical radius $r_{opt}$ with depth $z$ as follows (Brucker et al., 2010; Bingham and Drinkwater, 2000):

$$r_{opt}^n(z) = r_{top}^n + Q_n z \tag{4}$$

where $r_{top}$ is the optical radius at the surface in m, $Q_n$ is the snow grain-size gradient in m$^n$m$^{-1}$, and $n$ is the growth exponent. This model assumes a linear increase in optical radius ($n=1$), surface ($n=2$), or volume ($n=3$) with time from metamorphism, and therefore with depth assuming that snow accumulation is constant. It was shown by Brucker et al. (2010) to reproduce well the emissivities at 19 and 37 GHz within the Antarctic snow cover, and to be consistent with relative grain sizes over different regions of Antarctica.

**Surface ice crust:** The wind-glazed regions are covered by an ice crust formed by snow ablation from wind and sublimation (Courville et al., 2007). The thickness of this crust is likely millimetric, but there are other similarly thin ice layers embedded within the top few meters of the snowpack (Albert et al., 2004). Exact simulation of the thickness or depth of these ice layers is irrelevant because it is highly variable over the 12.5 km resolution used herein. We therefore model the ice crust as a single

| Parameter | Symbol | Unit | Values | Reference |
|---|---|---|---|---|
| Ice crust thickness | $z_{crust}$ | mm | 0 (no crust) to 10 | Albert et al. (2004); Courville et al. (2007) |
| Surface density | $\rho_{top}$ | kg.m$^{-3}$ | 300 to 450 | Albert et al. (2004); Leduc-Leballeur et al. (2015) |
| Surface optical radius | $r_{top}$ | mm | 0.1 to 0.5 | Courville et al. (2007); Brucker et al. (2010) |
| Density exponent factor | $B$ | m$^{-1}$ | 0.016 to 0.017 | Bingham and Drinkwater (2000); Leduc-Leballeur et al. (2015) |
| Snow grain-size gradient | $Q_1$ | mm.m$^{-1}$ | 0.005 to 0.1 | |
| | $Q_2$ | mm$^2$.m$^{-1}$ | 0.02 to 0.1 | Brucker et al. (2010) |
| | $Q_3$ | mm$^3$.m$^{-1}$ | 0.01 to 0.1 | |
| Temperature | $T$ | °C | -40 to -50 | Picard et al. (2022b) |
| Polydispersity | K | | 0.62 | Picard et al. (2022b) |

**Table 3.** Snowpack parameters for Antarctica SMRT simulations

high-density (917 kg.m$^{-3}$) ice layer on top of the snowpack, of thickness varying from 1 to 10 mm. Some of these thicknesses are smaller than $\lambda/4$ for ASCAT waves and the longest AMSR2 wavelengths. The SMRT can accommodate a single layer thinner than $\lambda/4$ by calculating explicitly the coherent effect through this layer following Montpetit et al. (2013); Proksch et al. (2015). This is implemented by setting the option *process_coherent_layer=True*, but can only process coherent scattering due to the layer thickness, not due to grain size.

The simulated snowpack therefore has six parameters, summarized in Table 3. The resulting depth and density profiles are shown in Fig. 4.

The SMRT model outputs the simulated brightness temperatures $T_B$ at each AMSR2 frequency in H and V polarizations. The emissivity is found by dividing the simulated $T_B$ by the assumed snowpack temperature. The model also outputs the simulated normalized radar cross-section $\sigma^0$ at each sensed polarization and incidence angle for QuikSCAT and OSCAT (see Table 1), and for ASCAT in VV polarization at incidence angles of 30°, 40°, 50°, and 60°.

## 4 Results and interpretation

### 4.1 Uniform density or grain size

We first tested the model with either density or optical radius (grain size) uniform with depth, while the other of the two varies as described above (Section 3.2). The result of the simulation for AMSR2 V-polarized emissivities is shown in Fig. 5 for physically reasonable values of the parameter values obtained by empirical trial-and-error; alternative values generally produce poorer fits. As shown in this figure, the model is unable to simultaneously fit all AMSR2 frequencies if either the optical radius or the density are kept constant.

Attempting to reproduce the observations with either the density or the optical radius uniform with depth is unsuccessful. If the optical radius is kept uniform, then we generally find excessive scattering (low emissivities) at high frequencies and insufficient scattering (high emissivities) at low frequencies (Fig. 5, left). This demonstrates that optical radius must increase

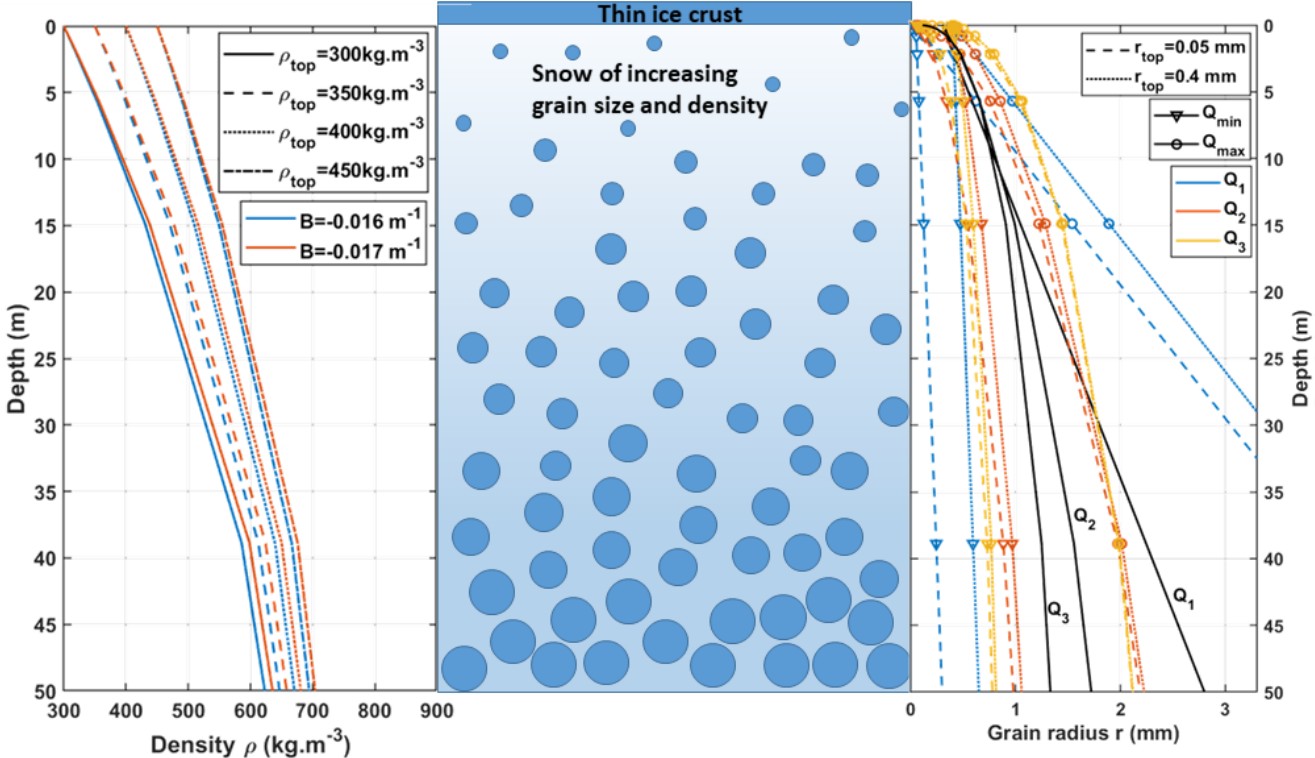

**Figure 4.** Schematic representation of the modeled snowpack, with (left) the density and (right) the optical radius variations with depth. Different line patterns correspond to different parameter values. The snow grains are shown here as circles for simplicity, but are not spherical neither in reality nor in the model; the complex grain shape is accounted for by the polydispersity $K$. On the optical radius plot, the profiles which reproduce the best the simulations are shown for $Q_1$, $Q_2$, and $Q_3$ in black; note that these three profiles are very similar in the top 15 meters, to which the observations are most sensitive.

with depth to fit the observed spectra. The increase in optical radius with depth in the Antarctica snowpack, as well as its influence on microwave emissivities, have long been known (e.g. Jay Zwally, 1977; Brucker et al., 2010). If instead the density is kept uniform, the simulated emissivities as a function of frequency have a concave shape, whereas a convex one is expected:

the lowest and highest frequencies appear to scatter excessively (Fig. 5, right). This would not occur if the density was not constant, consistent with increasing density with depth.

     Brucker et al. (2010) modeled the Antarctic snowpack at 19.3 and 37 GHz with increasing optical radius but constant density and found a reasonable match to the data (i.e. a roughly flat emissivity spectrum) everywhere in the dry zone. We are also able to reproduce the data in the megadune area with the same grain size profile equation at the same frequency range,

although with different values for the parameters $r_{top}$ (we find $r_{top} \approx 0.2$ mm; they find $0.45 < r_{top} < 0.65$ mm) and $Q_2$ (we find $Q_2 \approx 0.06$ mm$^2$.m$^{-1}$; they find $0.40 < Q_2 < 0.82$ mm$^2$.m$^{-1}$). This discrepancy is due to modeling differences between our simulations and those of Brucker et al. (2010). They used the DMRT-ML model composed of non-sticky spheres, whereas

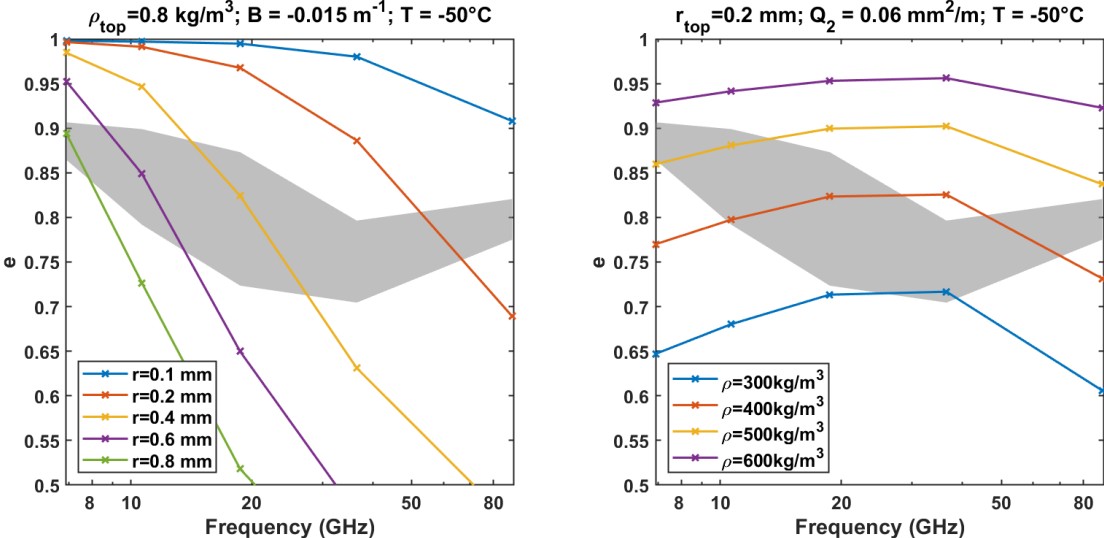

**Figure 5.** AMSR2 V-polarized emissivities simulated by the SMRT for a snowpack with either (left) uniform optical radius with depth or (right) uniform density with depth. The grayed region shows the full range of observed values within the ROI. The parameter values used for each simulation are indicated in the title.

we use the SymSCE model while accounting for polydispersity (as defined by Picard et al., 2022a). In any case, we show that while assuming constant density can be reasonable at two frequencies (19.3 to 37 GHz), our results demonstrate that it is inconsistent with observations at several frequencies over a wider spectral range.

### 4.2  Variable density and optical radius, without an ice crust

We then vary simultaneously the optical radius and the density with depth, as described in Section 3.2, but without any icy crust at the surface. Our goal is not to perfectly fit the model to the data, but instead to simulate the correct range of brightness temperatures and backscatters across all considered frequencies and polarizations, with the simplest possible snowpack model. For simplicity, we only show in Fig. 6 the combination of parameters which best matches the observations, for linear ($Q_1$), square ($Q_2$), and cubic ($Q_3$) increase in optical radius with depth. We plot emissivity in each polarization $e_V$ and $e_H$ versus frequency and $\sigma^0$ versus incidence angle in C-band (ASCAT) and in Ku-band (QuikSCAT and OSCAT).

We find that all considered microwave active and passive observations, except H-polarized emissivities, can be simultaneously simulated by the SMRT with reasonable parameter values. A summarized analysis of the influence of each parameter is given below.

Within the ranges expected in the Antarctic megadunes region, the snowpack **temperature** $T$ and the **density e-folding factor** $B$ have very little influence on both the emissivities and the $\sigma^0$ values ($\Delta e < 0.05$ and $\Delta\sigma^0 < 0.6$ dB). To simplify the visualization, simulated values are shown only for $B = -0.017$ m$^{-1}$, the value deduced from fitting two 80-m-long density

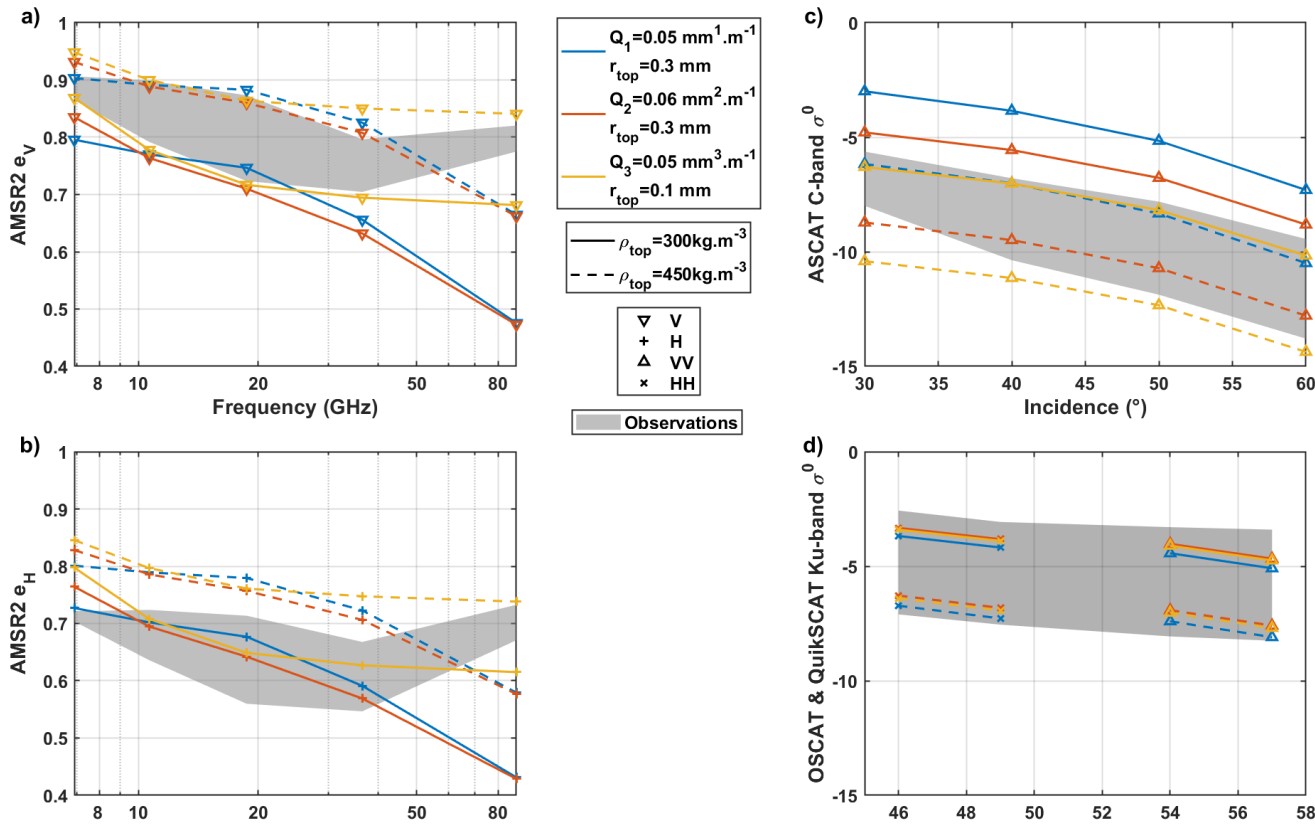

**Figure 6.** Best fitting simulations of AMSR2 V-polarized (a) and H-polarized (b) emissivity spectra and C-band ASCAT (c) and Ku-band QuikSCAT and OSCAT (d) backscatter $\sigma^0$ for different surface densities $\rho_{top}$ and optical radius profiles (gradient $Q_n$ and surface optical radius $r_{top}$), while fixing the temperature $T = -50°C$ and density e-folding factor $B = 0.017$ m$^{-1}$. For each set of data, the range of observations is shaded in gray. For simplicity, only the minimum and maximum values of density tested are shown: varying the surface density therefore corresponds to the space between the two lines of a given color. The range between the yellow lines (for $Q_3$) thus results in good fits for all, except H-polarized data.

profiles at dome C by Leduc-Leballeur et al. (2015), and $T = -50°C$, the annual mean temperature measured by Picard et al. (2022a) in the Antarctic megadunes.

The **grain size**, defined by the parameters $r_{top}$ and $Q_n$, is the primary control on the shape of the microwave emissivity spectrum. The surface optical radius $r_{top}$ affects high frequencies (which probe shallower depths) the most, whereas $Q_n$ controls the optical radius profile with depth and therefore influences lower frequencies. Assuming a linear increase of optical radius with depth ($Q_1$) leads to high optical radius at depth and small at the surface; ASCAT data can not be reproduced for the same configuration as AMSR2 data. In contrast, the cubic increase ($Q_3$) and square increase ($Q_2$) can simulate all data except H-polarized emissivities, though the square option ($Q_2$) also has difficulties with the 89 GHz simulations (see Fig. 6). As found by Brucker et al. (2010), we can thus exclude the case $n = 1$, but both other options are acceptable approximations: optical radius in the subsurface is probably a more complex function of precipitation rates, temperature, and random variations. The optical radii we find in the subsurface are very large, with values around 1 mm at 15 m depths for $Q_2$, and 0.9 mm for $Q_3$.

The **surface density** $\rho_{top}$ affects the backscatter and emissivity at all frequencies almost equally (Fig. 6). The density controls the effective dielectric constant of the medium, and therefore strongly affects the probed depths (with a denser medium being less transparent). Thus, to first order, the lower the density, the larger the path length within the medium, and the more opportunities for multiple subsurface scattering, leading to lower emissivities and higher $\sigma^0$. Scattering also varies in a complex manner with density (figure 2 of Picard et al. (2022b)); however, in our case this effect is difficult to disentangle from that of grain size, since both properties vary simultaneously with depth.

We find that the **H-polarized AMSR2 emissivities**, shown in Fig. 6b, are poorly reproduced by the parameters that match the rest of the data best, and would require higher optical radius values ($r_{top}$ and/or $Q_2$) than the V-polarized emissivities. This is likely due to the polarization properties of the medium, caused by layers of varying density (embedded thin ice layers) and by a surface crust. Indeed, ice layers remain mostly invisible to V-polarized radiation near the Brewster angle, whereas H-polarized radiation is very sensitive to these vertical dielectric contrasts (Leduc-Leballeur et al., 2015). Radar observations in HH polarization should be equally affected, yet the simulations fall in the observed range in Fig. 6b. We do not have an explanation for this different behavior.

The simulated **89 GHz emissivity** ($\lambda = 3.34$ mm) in V polarization is often very low, especially for surface optical radius values $r_{top} \geq 0.2$ mm. These large grains (compared to the wavelength) cause very important simulated scattering at 89 GHz. The $n = 3$ option allows to start with very low optical radius at the surface and increase quickly with depth; yet even so, the 89-37 GHz slope is never reproduced (Fig. 6). At the depths probed at 89 GHz (a few centimeters), the simulated snowpack is in fact not realistic enough. Indeed, it does not account for the observed ice crust in the wind-glazed regions and the stochastic variations of density and optical radius at these depths that may have not been averaged using a single year of observations (Stefanini et al., 2024).

### 4.3 With an ice crust

To attempt to improve the fit the H-polarized AMSR2 emissivities and the 89 GHz V-polarized AMSR2 emissivities, we test adding an ice crust of variable thickness on top of the snowpack. Such an ice crust is common within the wind-glazed regions

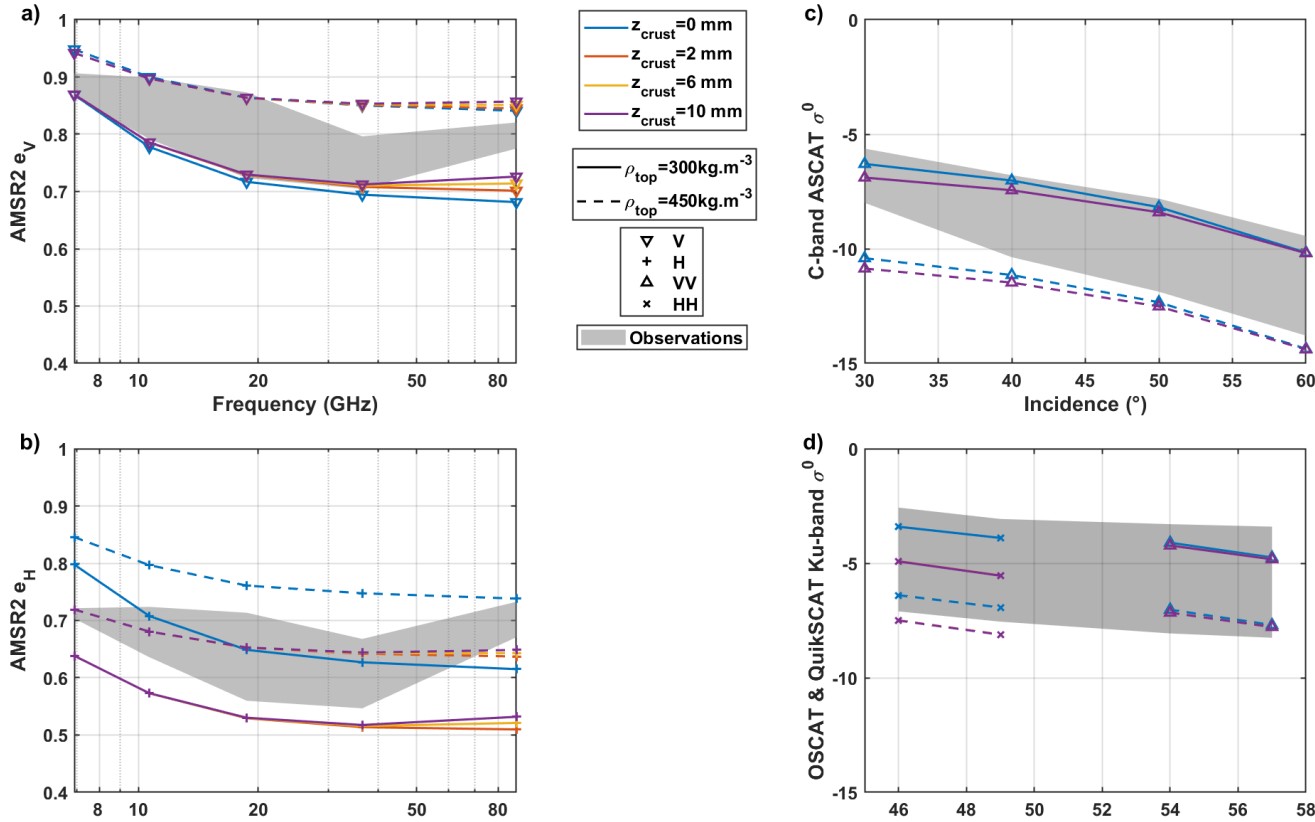

**Figure 7.** AMSR2 V-polarized (a) and H-polarized (b) emissivity spectra and C-band ASCAT (c) and Ku-band QuikSCAT and OSCAT (d) backscatter $\sigma^0$ for different values of the surface density $\rho_{top}$ and ice crust thickness $z_{crust}$ while fixing the surface optical radius $r_{top} = 0.1$ mm , grain-size gradient $Q_3 = 0.05$ mm$^3$.m$^{-1}$, temperature $T = -50\,°$C, and density e-folding factor $B = 0.017$ m$^{-1}$. As in previous plots, the full range of observed emissivities and backscatter values within the region of interest is shaded in gray. The simulated scatterometry is identical for $z_{crust} = 2, 6$, or 10 mm, and overlap in c) and d). Note that, for $z_{crust} = 0$ (no crust), the simulated values are the same as highlighted in orange in the previous section.

and is likely to polarize the outgoing signal, decreasing H-polarized emissivities. The results of these simulations are shown for
335 $e_V$, $e_H$ , C-band $\sigma^0$, and Ku-band $\sigma^0$ (Fig. 7), for $T = -50$°C, $B = -0.017$ m$^{-1}$, $r_{top} = 0.1$ mm, and $Q_3 = 0.05$ mm$^3$.m$^{-1}$. These parameter values, which reproduce reasonably well the expected $\sigma^0$ and $e_V$ without an ice crust, were chosen for the figure, but the effect of the ice crust is similar for all other tested parameter combinations.

   **V polarized emissivities** $e_V$**:** As shown in Fig. 7a, an ice crust $\leq 1$ cm thick can change the 89 GHz emissivities by at most 0.04, and remains insufficient to explain the observed 37-89 GHz slope. A thicker crust might be able to reproduce the
340 observations, but would not be realistic: the thickness of the ice crust and underlying ice layers observed in the field is of the order of millimeters, not centimeters. Here, our simplified model seems insufficient, and it may be necessary to add complexity

such as superimposed layers of randomly varying density to improve the fit. At 89 GHz, radiometry is also very sensitive to surface temperature, which is seasonally much more variable than the temperature at depth.

**H polarized emissivities** $e_H$**:** Comparing the emissivity spectra for $z_{crust} = 0$ (no ice crust) with the others (all identical below 37 GHz), Fig. 7b shows that the polarization induced by the ice crust decreases $e_H$. This decrease is sufficient, and even slightly in excess, to reproduce the H-polarized observations. This is true regardless of the thickness of the ice crust, because the air-surface interface, which controls the H polarization, is the same.

**Scatterometry** $\sigma^0$**:** The ice crust decreases the simulated QuikSCAT and OSCAT backscatter in HH polarization, but does not affect the VV-polarized scatterometry (Fig. 7c and d). Regardless of the crust thickness, the effect on the HH-polarization is so strong in the simulations that $\sigma^0$ is predicted to be lower at 46° and 49° incidence in HH than at 54° and 57° in VV. This does not match the OSCAT and QuikSCAT observations, which show $\sigma^0$ decreasing with incidence angle regardless of the polarization (Fig. 3). This could perhaps be explained by the presence of surface roughness, regional heterogeneity within the 12.5 km pixels, or coherent scattering, which can affect the polarization ratio of active data (Hofgartner and Hand, 2023).

Including a thin ice crust in SMRT simulations helps lower the H-polarized emissivities, but is insufficient to explain the 89 GHz $e_V$ and $e_H$, and even introduces new discrepancies to the H-polarized scatterometry. Thus, a simple snowpack model is sufficient to reproduce the 6.9–37 GHz $e_V$ and 5.2–13.5 GHz $\sigma^0$, but more complexity must be introduced for higher frequencies and for H polarization. This complexity can include a thin ice layer as tested here, but also a time- and depth-varying temperature profile, random layer densities and optical radii, a different optical radius gradient in the top few centimeters, or surface roughness. While this result illustrates the limits of a simple snowpack model, it also highlights the richness of these complementary datasets, which used in synergy can provide constraints on subsurface properties inaccessible from one frequency, polarization, or mode alone.

## 5 Discussion

### 5.1 Wind-glazed regions of Antarctica as an analog for Saturn's icy moons

Antarctic wind-glazed regions can serve as analogs for icy moon surfaces. Their consistently low temperatures prevent melting, and their high-purity ice, minimal precipitation, and near-zero accumulation resemble the nearly atmosphere-less conditions of moons like Enceladus. Additionally, abundant satellite data and some in situ measurements provide strong constraints for testing modeling approaches before applying them to icy moons.

Yet, like all Earth analogs for other planetary surfaces, this one is also imperfect. The temperature remains much higher than in the outer solar system, where airless bodies can also witness large surface temperature variations (e.g., between 40 and 140 K for Saturn's moons, Howett et al., 2010). The higher Earth temperatures lead to faster sublimation and development of depth hoar, as well as slightly different dielectric properties and much higher absorption losses: microwaves can probe considerably deeper in colder ice. The second major difference is the presence of an atmosphere, with important katabatic winds (Scambos et al., 2012) and occasional precipitation. Meanwhile, icy moons are airless but are affected by exogenic processes, including

impact gardening and electron bombardment, likely causing differences in the surface structure (crystal size, presence of a crust, depth of regolith...), which remains poorly understood.

The introduction of a new icy moon analog is encouraging for further work analyzing the microwave properties of icy moons. It could be used to examine the role of coherent backscatter in Earth snow (Stefko et al., 2022) or the variations of microwave signals with the presence of non-icy material even in small quantities, which has been observed on icy moons (Le Gall et al., 2019, 2023).

## 5.2 Implications for Saturn's icy moons

We have seen that the SMRT is able to successfully simulate simultaneously active and passive Ku-band observations in the Antarctica megadunes region (Section 4). Our main objective now is to check if a simple configuration of the SMRT is also able to reproduce at the same time the Ku-band radiometry and radar observations of Saturn's mid-sized icy moons Mimas, Enceladus, Tethys, Dione, Rhea, Iapetus, and Phoebe by the Cassini radar (Le Gall et al., 2019, 2023). We therefore chose a simple two-layer model of regolith (modeled as snow in the SMRT) on top of ice, with a constant temperature profile, at the Cassini radar frequency (13.78 GHz), and with the default dielectric properties from Matzler et al. (2006). There are thus six parameters as shown in Table 4: the density and optical radius of each layer, the thickness of the regolith, and the temperature. Given the large uncertainties and likely large inter- and intra-satellite variations in the density, grain size, composition, and structure of icy moon regoliths, each parameter is left to vary within a very large range.

The regolith of icy moons at meter depths is poorly understood, with some parameters entirely unconstrained. Measured surface porosities vary depending on the satellite, model, and data from 0.05 to 0.99 (Carvano et al., 2007, and references therein), but generally need to be very high to match near to far infrared observations (Carvano et al., 2007; Ciarniello et al., 2011; Ito et al., 2022). Compaction due to gravity is insignificant for hundreds of meters to tens of kilometers of depth, especially on Saturn's mid-sized moons (Mergny and Schmidt, 2024), but other processes such as cryovolcanic or tectonic activity, impacts, radiation, or sintering could affect both density and grain size (Molaro et al., 2019). Grain size is similarly poorly constrained, with surface values measured between 1 $\mu$m (Ito et al., 2022), tens of $\mu$m, and up to 200 $\mu$m in the Enceladus South polar terrain (Jaumann et al., 2008; Taffin et al., 2012); subsurface values remain unknown. Grain radii beyond 1 mm are possible, but are not simulated herein due to the transition to a Mie scattering regime, which is not included in the SMRT model. Grain radii below 50 $\mu$m are too small to significantly affect the scattering at 2.2 cm wavelength, so they are not tested in the model. The temperature can vary from about 40 to 140 K at the surface (Howett et al., 2010), but is less extreme in the subsurface probed by microwaves (Bonnefoy et al., 2020; Le Gall et al., 2023): we therefore chose a range of 60 to 120 K, which is still very wide. Finally, the thickness of the porous regolith is generally assumed to be of the order of meters (Bland et al., 2015; Ries and Janssen, 2015), but is of the order of hundreds of meters at least near the pit chains of Enceladus (Martin et al., 2023). These various observations justify the large parameter ranges shown in Table 4. Meanwhile, polydispersity is assumed to be the same as in Antarctica. Grain radii are kept below 1 mm because beyond that value, Mie scattering, which is not accounted for the SMRT, becomes significant in Ku band (2.2 cm for the Cassini Radar).

| Parameter | Symbol | Unit | Values | Comments and reference |
|---|---|---|---|---|
| Regolith grain radius | $r_{top}$ | mm | 0.05 to 1 | Jaumann et al. (2008); Taffin et al. (2012); Ito et al. (2022) |
| Regolith density | $\rho_{top}$ | kg.m$^{-3}$ | 10 to 450 | Carvano et al. (2007); Ciarniello et al. (2011); Ito et al. (2022) |
| Ice optical radius | $r_{ice}$ | mm | 0.05 to 1 | |
| Ice density | $\rho_{ice}$ | kg.m$^{-3}$ | 500 to 900 | |
| Regolith thickness | $Z_{top}$ | m | 1 to 1000 | Bland et al. (2015); Martin et al. (2023) |
| Temperature | $T$ | K | 60 to 120 | Howett et al. (2010); Le Gall et al. (2023) |
| Polydispersity | $K$ | | 0.62 | |

**Table 4.** Snowpack parameters for icy moon SMRT simulations

Most observations of the icy moons by the Cassini radar / radiometer were not resolved, and included multiple incidence angles simultaneously. We therefore request the SMRT to output the normalized radar cross-section $\sigma^0$ and the brightness temperature $T_B$ in both polarizations (H and V for radiometry; HH and VV for radar) and for all incidence angles with a step of 5°. Emissivities $e_V$ and $e_H$ are calculated by dividing the brightness temperature by the assumed medium temperature $T$. Both $\sigma^0$ and $e$ are then integrated over the disk (only in same-sense-linear polarization for $\sigma^0$), following Wye et al. (2007) and Le Gall et al. (2019) while also accounting for both polarizations:

$$A_{SL}^{disk} = \int\limits_{0}^{\pi/2}\int\limits_{0}^{2\pi} (\sigma_{VV}^0(\theta,\phi)\cos^2\phi + \sigma_{HH}^0(\theta,\phi)\sin^2\phi)\sin\theta\, d\phi\, d\theta /\pi \tag{5}$$

$$e^{disk} = \int\limits_{0}^{\pi/2}\int\limits_{0}^{2\pi} (e_V(\theta,\phi)\cos^2\phi + e_H(\theta,\phi)\sin^2\phi)\sin\theta\cos\theta\, d\phi\, d\theta /\pi \tag{6}$$

The resulting disk-integrated values are then directly comparable to those observed by the Cassini radar / radiometer and examined by Le Gall et al. (2023). For comparison, we also simulate equivalent disk-integrated observations for the Antarctica megadunes region. Using the same SMRT configuration as in Section 3 with our best-fitting parameters ($T = -50°$C, $B = -0.017\,\mathrm{m}^{-1}$, $r_{top} = 0.1$ mm, $Q_3 = 0.05\,\mathrm{mm}^2.\mathrm{m}^{-1}$, and $300\,\mathrm{g.m}^{-3} < \rho_{top} < 450\,\mathrm{g.m}^{-3}$), we simulate Ku-band observations at all incidence angles and calculate $A_{SL}^{disk}$ and $e^{disk}$ as above.

Plotting all disk-integrated values simulated by the SMRT regardless of the parameters over the Jupiter and Saturn moon observations from Le Gall et al. (2023) shows that the model can reproduce observations of Jupiter's icy moons, but never of Saturn's moons, which remain too radar-bright (Fig. 8). We hypothesize that this is caused by the model simulating only incoherent scattering, whereas the coherent backscatter opposition effect (CBOE) has been hypothesized to be the main cause for icy moons unusual radar properties (Black, 2001; Le Gall et al., 2019, 2023; Hofgartner and Hand, 2023). In theory, the CBOE can at most multiply the radar returns ($\sigma^0$ and therefore also $A^{disk}$) by a factor of 2. To investigate the upper limit of SMRT simulations if they were including CBOE, we plot the simulations multiplied by 2, and find that these are consistent

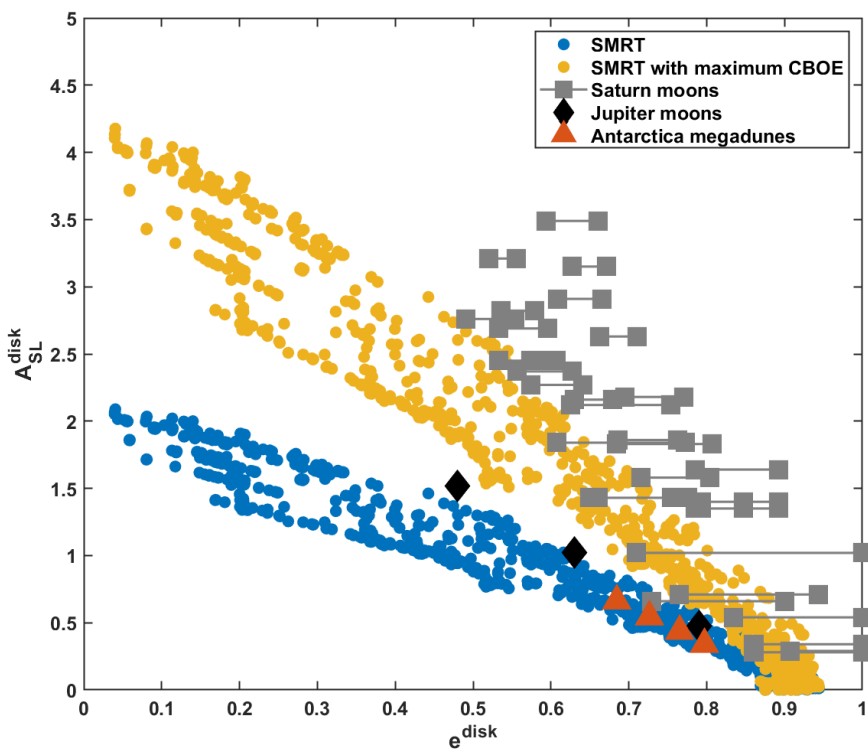

**Figure 8.** The results of SMRT simulations for a two-layer icy satellite model are shown, along with observations on Jupiter's and Saturn's icy satellites (from Le Gall et al., 2023) and simulated disk-integrated observations corresponding to the Antarctica megadunes region. Since the SMRT does not simulate the coherent backscatter opposition effect, $A_{SL}^{disk}$ is multiplied by 2 to provide the upper theoretical limit of the simulations with the maximum possible COEB effect. Even so, the model is unable to reproduce most Cassini observations of Saturn's icy moons.

with some Saturn moon observations, but still insufficient for Enceladus, the radar-brightest object in the solar system. This exercise reinforces the hypothesize that CBOE is necessary to explain moon backscatter, although it may not fully explain the behavior of the brightest moons. Integrating CBOE into SMRT is a meaningful avenue for future improvement.

## 6 Conclusions

We use the SMRT model to simulate AMSR2 6.9 to 89 GHz emissivities as well as C- and Ku-band scatterometry measured by ASCAT, OSCAT, and QuikSCAT in the Antarctic megadunes region. Modeling the Antarctic snowpack as layers with increasing density and grain size with depth allows us to reproduce simultaneously, for the first time, all V-polarized emissivities from 6.9 to 37 GHz and backscatter at 5.2 and 13.4 GHz. The observed microwave emissivity spectrum is slightly convex, but

remains within the range $0.7 < e_V < 0.9$: this shape can only be explained with both density and grain size increasing with depth, which result from densification and metamorphism, respectively. Combining multiple frequencies thus provides insight into the vertical profile, an approach which is also useful on icy satellites such as Iapetus (Bonnefoy et al., 2024) and Ganymede (Zhang et al., 2023; Brown et al., 2023), although variations in composition will also have to be considered.

The fact that 5–37 GHz Antarctica radar and radiometry data can be successfully simulated while ignoring coherent scattering, surface roughness, and random density and grain size fluctuations in the top few meters implies that both the active and passive V-polarized microwave signals at these frequencies are dominated by incoherent scattering on subsurface grains. Meanwhile, to reproduce H-polarized emissivities, we found that adding an ice crust over the snowpack, as observed in the wind-glazed regions, is a viable configuration; however it also reduces the HH-polarized scatterometry. This could likely be

remedied by adding further complexity to the model, such as surface roughness which tends to moderate the polarization effect of flat interfaces. It is also possible that coherent scattering, either on large grains (Hapke, 1990) or caused by interference between layers (Leduc-Leballeur et al., 2015), plays a role in the scatterometry data. A strong difference between H and V (or HH and VV) is an indicator of a stratified snowpacks, or the presence of pure ice at the surface. Similarly, the observed 37–89 GHz slope in V- and H-polarized emissivities cannot be reproduced by the model, even when including a thin ice crust. A more

realistic near-surface model appears to be required for the 89 GHz data, for instance with a seasonally varying temperature, or by modeling the strong gradient in grain size in the topmost 10-20 cm of the snowpack as observed in the field (Picard et al., 2022a).

The relative success of the simple snowpack representation and of the SMRT in reproducing 5–37 GHz active and passive microwave observations in the Antarctic megadunes region encourages its application to the radar and radiometry observations

on icy moons, characterized by even higher scattering. A first application to icy moons using a wide range of parameters on a two-layer model is able to reproduce observations of Jupiter's moons. Meanwhile, to approach the observed radar and radiometry data on Saturn's moons, it is necessary to invoke coherent backscattering, consistent with previous work (Hofgartner and Hand, 2023). Even so, Saturn's satellites and especially Enceladus remain too radar-bright for the model configuration and range of parameters tested, hinting for example at very large grain sizes. The SMRT, with realistic hypotheses, thus significantly

improves upon simulations by previous analytical models (Le Gall et al., 2023), but also highlights the importance of coherent scattering, which future work could incorporate into the model, for instance using the formulations of Markkanen and Penttilä (2023) or Muinonen and Penttilä (2024). CBOE is also likely on Earth (Stefko et al., 2022): successfully understanding, measuring, and modeling it will benefit both communities. It would also be interesting to attempt to reproduce resolved observations of Saturn's moons using more constraints such as resolved observations (Le Gall et al., 2014, 2017; Bonnefoy et al.,

2020) or radiometry at multiple frequencies (Bonnefoy et al., 2024), for instance on Enceladus or Iapetus. Meanwhile, integrating non-icy components into the model may help understand both regional and satellite-to-satellite variations in microwave emissivity and radar backscatter.

## 7 Environmentally responsible research

The work presented herein makes use of data from four Earth observation space missions: QuikSCAT, whose launch mass was 970 kg; ASCAT on the Metop-C satellite, whose launch mass was 3950 kg; OSCAT on Scatsat-1, whose launch mass was 377 kg; and AMSR2 aboard GCOM, whose launch mass was 2000 kg. Using a life-cycle emission factor of $50(\pm10)$ $tCO2e.kg^{-1}$ (Wilson, 2019; Knödlseder et al., 2022; Marc et al., 2024), these satellites have estimated carbon footprints of, respectively, $18,500$ $tCO2e$, $197,500$ $tCO2e$, $18,850$ $tCO2e$, and $100,000$ $tCO2e$, without accounting for other environmental impacts such as local pollution due to mining or ozone layer depletion caused by the launches. Since data from these satellites are not used only for research purposes but also for weather forecasting and private applications, it is difficult to estimate the impact per scientific paper. We also made use of data from the Cassini mission, which using the same method had an estimated carbon footprint of $392,902$ $tCO2e$, corresponding to $42$ $tCO2e$ per paper as of 2022 (Knödlseder et al., 2022). We wish to warn against unnecessary proliferation of infrastructures with such high environmental impact and instead promote sufficiency in both Earth and Planetary Science.

*Code availability.* SMRT used in this submitted manuscript is available from https://github.com/smrt-model/smrt.

*Author contributions.* LEB ran the simulations and wrote the manuscript. All authors read and commented on the manuscript.

*Competing interests.* The authors declare no competing interests.

*Acknowledgements.* G.P. was supported by the French Polar Institute (IPEV) project number 1110. We thank the editor and two anonymous reviewers for their valuable comments that helped improve the manuscript.

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
