# Peer review of "Active-passive microwave scattering in the Antarctica wind-glazed region: an analog for icy moons of Saturn"

_EGUsphere, 2024_

## Referee Comment (RC2)

Review comments on TC paper 2024-3972 Active-passive microwave scattering in the Antarctica wind-glazed region: an analog for icy moons of Saturn

This work provides an active and passive simulation study for the East Antarctica using SMRT model. Simulations are done for a wide range of frequency channels from 5.2 to 89GHz. The authors wants to draw an analogy between the ice moons and this particular region of Antarctica and looks like the authors want to claim that this region would be a good example for the study of icy moons.

From my personal perspective, some major points need to added to the paper and some concerns need to be resolved before the paper can be published.

Here are some general comments for the paper:

1. In the abstract, the authors need to provide some conclusions that they obtained from this study and also need to provide the "up-shot"(how would this study contribute to a "larger picture" and would help answer a problem).
2. If the goal of the paper is to show that the Region of Interest (ROI) in the East Antarctica is a good analogy for icy moons, the active and passive data signature, and the measurement set-up for the icy moons needs to be presented and the features of the icy moons and ROI needs to be discussed. In such a way, the analogy could be drawn. Currently, the discussions are not sufficient.
3. Since the paper is majorly doing simulation to match up the observations, if parameters from icy moons can reproduce the measurements over ROI, this can also imply an analogy.

Detail comments are the following:

1. Resolution. As indicated by the sensor parameters, the scatterometers and radiometers are having different resolutions(ASCAT, Qscat 25km, AMSR2 based on frequency). In this work, the authors project the different data sets into uniform 12.5km grids. In such a way, the near by data pixels would be highly correlated and would not provide extra information for pixels within the resolution of a given data set. Such a interpolation would ignore the heterogeneity within a large resolution and may mistakenly use the coarse, larger area averaged measurement to represent the measurement for a smaller area. I believe a better way is to aggregate the high resolution data into low resolution such that different data sets can have the same averaging effect over the measured area. Can the authors provide some discussion on this?
2. If my memory serves me correctly, L3 data from AMSR is already grided. That data set might be better? Only a suggestion.

3. The way of data averaging is not clear to me. How is the measured data averaged to a data point in each frequency?

4. In matching the data, active part looks fine to me, but the passive part doesn't look satisfactory. The observables from radiometers are brightness temperatures, emissivity values are derived values. Radiometers are very accurate, usually the errors are within 3K, assuming a physical temperature of 270K, this corresponds to an error in emissivity around 0.011. I would suggest the authors show the comparison in terms of brightness temperature. In such a way, the forward simulation would show a difference of 10K or more. Match up can be improved.

---

## Author Comment (AC1)

**Response to Reviewer 1**

Summary:

This paper presents simulations of both active and passive microwave observations over the Antarctic wind-glazed region using the SMRT (Snow Microwave Radiative Transfer) model across frequencies from 5 to 89 GHz. The study aims to evaluate the Antarctic as an Earth analog for Jupiter and Saturn's icy moons, and applies the same modeling approach to explore the radiative properties of icy moon surfaces. The results show promising potential in using Antarctic conditions to interpret extraterrestrial scattering environments, and highlight SMRT's flexibility in handling both passive and active configurations.

However, in my opinion several issues require clarification before publication:
The description of the scatterer configuration requires clarification and refinement. In particular, terminology such as "microwave grain size," "optical grain size," and "stickiness" should be more clearly defined. These terms are often used interchangeably or without context, which creates confusion. It would greatly benefit the reader if the authors could explain how these terms are derived, how they relate to measurable physical properties (e.g., grain radius, correlation length, microstructural anisotropy). A more precise treatment of these definitions would strengthen both physical interpretation and modeling transparency.

We thank the reviewer for a careful and thorough reading of our manuscript, as well as for the constructive suggestions that have significantly helped improve the quality and clarity of the paper.

We fully agree with the necessity of rigor in the definition of the scattering parameters. We propose to update the definitions in Section 3.2 and add references:

- The microwave grain size is defined in Eq 2 and the reference is present.
- The Porod length "lp" is used in Eq 2, and was defined in Eq 3 but with the wrong symbol "lc". We corrected this typo.
- The optical radius is used in Eq 3, and its definition is now added, along with a citation.
- We have changed many occurrences of "grain radius" or "grain size" to "optical radius". We have only kept "grain size" when we aim to designate the grain size in general.
- the term "stickiness" was mentioned as an example only, it is now removed, as it is not relevant for this paper (at least not as defined originally by Tsang et al.).

In addition, we propose the following changes in Section 3.1: we have reformulated a sentence that used polydispersity before its definition.
**"The snow microstructure model used for each layer is the scaled exponential as proposed in \citet{picard2022a} with parameters defined in Section \ref{sec:params}"**

The treatment of the Coherent Backscatter Opposition Effect (CBOE) correction and the relationship between grain size and wavelength require more thorough justification, particularly in light of the frequency-dependent scattering regimes involved.

We do not intend to provide an exact or comprehensive treatment of the Coherent Backscatter Opposition Effect (CBOE), given the well-known difficulties highlighted both in the manuscript and in the reviewer's detailed comments (see L.398). Instead, our objective is to test the hypothesis that CBOE could potentially account for the passive and active observations of Saturn, as has already been suggested in the literature. Accordingly, the paragraph addressing CBOE has been revised to improve clarity and to better reflect the scope of our analysis. We also refer the reviewer to our specific response to comment L.398 for further detail.

There appears to be inconsistency in the identification and justification of the best-fit model. If the simulation results are intended to represent best fits, the retrieval methodology and whether a formal optimization was performed should be explicitly described.

Our goal is not to find a best fit solution. Rather, our objective is to investigate whether it is possible to identify a set of parameters that simultaneously yields a realistic range of brightness temperatures and backscatter values. The exact numerical values of the snowpack parameters are not, in themselves, critical for our purpose. What is essential is to assess whether a parameter combination exists that can simultaneously agree with the multiple satellite observations, thereby testing the plausibility of our modeling approach.

In this sense, the exercise should be understood as an exploration of the parameter space rather than as an attempt to derive unique or physically constrained solutions. Our central aim is to determine whether simulations based on a simple snowpack model can reproduce the observations consistently across all considered frequencies and modes (both active and passive).

Despite these issues, the paper offers valuable insight into SMRT's performance in cold, dry, high-frequency regimes, and contributes to understanding terrestrial–planetary analogs. Addressing the points above will significantly strengthen the scientific clarity and rigor of the study.

Thank you!

Specific comments:

Line 1-5:
The phrase "complimentary modes" should be corrected to **"complementary modes".**
Additionally, the expression "top meters to hundreds of meters" could be refined to "upper meters to several hundred meters" for improved readability.

Thank you. Change has been made to the text.

Line 5-10:
The repetition of "with" in "with a simplified snowpack with constant temperature" may be improved by rephrasing.

We propose **"The data are simulated using the Snow Microwave Radiative Transfer (SMRT) model, assuming a simplified snowpack characterized by constant temperature and a continuous increase in grain size (represented by optical radius) and density with depth."**

Line 15-20:
The use of "diffuse scattering" in this sentence is slightly ambiguous and breaks the otherwise quantitative list structure. Consider rephrasing or elaborating to clarify whether the diffuse behavior refers to surface roughness, volume scattering, or both, and how it contributes to the anomalous radar response.

We propose to remove this term which does not add much information.

Line 20-25:
The phrase "simple radiative transfer models" is somewhat vague. It would strengthen the introduction to specify what assumptions or limitations define these models (e.g., homogeneous layering, absence of coherent effects, neglect of subsurface structure). This would help the reader understand the nature of the modeling challenge and the physical processes potentially being overlooked.
"Indeed, the Cassini Radar instrument, which operated in the Ku-band (13.78 GHz frequency, 2.2 cm wavelength) measured high backscatter but also higher than expected microwave emissivities."
The sentence as written is ambiguous. It is unclear whether the high backscatter and elevated emissivity are each individually unexpected, or whether it is their coexistence at the same frequency that is anomalous. Clarifying this would improve the reader's understanding of the observational paradox being described.

The term "simple" is indeed vague and inadequate. The important point is that the calculations were unsuccessful to simulate both active and passive microwave observations simultaneously.

The backscatter is unexpectedly high, but the corresponding emissivity is also rather high, compared to what would be expected from such high backscatter. We propose to modify the text as follows: "However, \citet{janssen2011, bonnefoy2020, legall2023} highlight another anomalous aspect of microwave scattering on Saturn's icy moons: radiative transfer **calculations** are unable to simultaneously explain active (radar) and passive (radiometry) observations at the same frequency. Indeed, the Cassini Radar instrument, which operated in the Ku-band (13.78 GHz frequency, 2.2 cm wavelength) measured **very high backscatter as well as higher than expected microwave emissivities (as compared to what would be modeled for a high backscattering surface). "**

Line 25-30:
"The surfaces of Saturn's icy moons are constituted primarily of high-purity water ice…"
It would be helpful for the authors to clarify whether this refers solely to the immediate optical surface (as seen in reflectance spectroscopy), or if it includes the shallow subsurface layers that contribute to radar and microwave responses.

This refers to both surface and subsurface icy moons. This has been clarified in the text.

'The surfaces **and sub-surfaces of Saturn's icy moons (as observed by visible to microwave instruments)** are constituted…'

Line 45-50:
"…which is the least emissive and most scattering region…"
The rationale for selecting a region described as "least emissive" warrants further clarification. Given that the icy moons of Jupiter and Saturn are characterized by both high microwave emissivity and strong radar backscatter, it is not immediately clear how a low-emissivity terrestrial region serves as an appropriate analog.
The phrase "transports and sublimates snow downslope" is ambiguous. It is unclear whether the authors intend to describe snow sublimation as occurring simultaneously with transport, or as a separate surface process enhanced by katabatic wind.

Our response to the comment Line 20-25 should address this issue. In addition, we propose to swap high scattering and least emissive, the first term being the main criteria for our selection. **"which is the most scattering and least emissive region of Antarctica".**

Line 50-55:
 "These large crystals, very scattering at microwave frequencies, are responsible for the observed low emissivities and high radar backscatter". The expression "very scattering" is informal and potentially ambiguous. Consider rephrasing to describe the crystals as strong or efficient microwave scatterers.

The sentence can be reformulated: "These large crystals are **strong and efficient scatterers at microwave wavelength. They** are responsible for the observed low emissivities and high radar backscatter \citep{fahnestock2000}. **"**

Line 55-60:
The phrase "cause important scattering" is imprecise. Consider replacing it with more specific terminology, such as "enhance subsurface volume scattering" or "increase microwave scattering efficiency." Additionally, you may wish to briefly clarify how sublimation–deposition cycles contribute to the formation of polygonal grains and grain chains.

We propose to merge this sentence with the previous one and to add a reference for the process of crystal growth **\citep{courville2007}**.

Line 60-65:
The use of "shorter frequencies" is not standard terminology. Consider revising to "lower frequencies"

This has been corrected.

The explanation of how increased penetration depth and enhanced scattering by larger grains "compensate each other" to yield a flat brightness temperature spectrum between 19 and 37 GHz lacks clarity. It may be useful to first clarify that lower frequencies (e.g., 19 GHz) penetrate deeper and may therefore sample warmer subsurface layers, potentially

increasing brightness temperatures. However, the statement that "longer wavelengths are also sensitive to larger grain sizes" is not always valid, especially considering that the wavelength difference between 19 GHz and 37 GHz is only a factor of two. It would be helpful to clarify the intent behind introducing this flat spectral behavior: is it meant to serve as evidence of grain size increasing with depth, or simply a result arising from known structural gradients? If it is intended as evidence of depth-dependent grain evolution, the current explanation is not sufficiently convincing. If it is instead presented as an outcome, then rephrasing to reflect that distinction would improve clarity and interpretation.

We propose the following reformulation:
**"The depth probed by a microwave radiometer being approximately proportional to the wavelength, longer wavelengths (i.e., lower frequencies) probe deeper, where grains are larger in size in Antarctica. The longer wavelengths are thus subject to stronger scattering relatively to the homogeneous snowpack, leading to lower emissivities. The shorter wavelengths are less affected by the vertical gradients. This leads to a flatter emissivity spectrum at 19 to 37 GHz as successfully modeled by \citet{brucker2010} using the dense-medium radiative transfer model multi-layer model (DMRT-ML) \citep{picard2013}."**

The temperature gradients in the snowpack are not considered in this paragraph because we intended to explain the spectrum of emissivity. This is now explicit. In the following, we work with mean annual temperature (and brightness temperature) which is almost equal at any depth (0-20m).

Line 86
The role of multi-frequency observations in constraining vertical variations in grain size and density is important, and could be more clearly articulated. It is not immediately clear what the authors mean by the "frequency dependence of grain size and density." Grain size and density are physical properties of the snowpack, not functions of frequency. Rather, it is the instrument's sensitivity to those properties that varies with frequency, due to differences in penetration depth and scattering behavior. Clarifying this distinction would strengthen the physical interpretation and aid in understanding how multi-frequency data constrain subsurface structure.
It would be helpful for the authors to include the spatial resolution of each observational dataset in Table 1.

We propose to reformulate using a more precise language:
'**Additional frequencies, in both active and passive modes, help to further constrain interpretations, clarify the spectral sensitivity to parameters such as grain size and density, and evaluate the model ability to successfully reproduce multi-frequency active and passive observations.**'

**The instrument spatial resolutions are added in Table 1.**

Line 95-100
The explanation of the Kohonen clustering could benefit from additional clarification. While it is mentioned that the algorithm identifies clusters with neighborhood constraints, it would

be useful to briefly describe what the clusters represent in the context of the dataset—i.e., distinct combinations of backscatter and emissivity properties or distinct physical parameters. Clarifying how the number of clusters (10) was chosen and what physical insights each group provides would help readers interpret Figures 2 and 3 more effectively.

The text has been modified, following the reviewer's recommendation:

'This unsupervised machine learning algorithm integrates the multi-dimensional nature of the passive and active observations, and groups the observations into clusters with similar features in the observation space. It synthetically describes the co-variability of the different observations. The clusters are self-organized with neighborhood requirements, meaning that when the classification has converged, nearby clusters have nearby characteristics in the observation space. The number of clusters is chosen so that at least one piece of observation (at one frequency, one polarization, and one mode) shows a clear statistical difference between clusters, i.e., the difference between clusters is greater than the standard deviation of that observation within these clusters. The classification in 10 clusters is shown in Figs. 2 and 3, whose comparison highlights the anticorrelation between active and passive data, with very radar-bright regions exhibiting low emissivities.'

Line 107
The use of ERA5 skin temperature to compute observed emissivity should be more carefully justified in light of the modeling assumptions. In the simulations, a uniform temperature profile is applied, whereas the skin temperature used for observed emissivity inherently contains seasonal and diurnal variability. This mismatch is especially important at lower microwave frequencies, which probe deeper into the snowpack where temperatures remain relatively stable and are decoupled from short-term surface fluctuations. The authors should clarify how the observationally derived emissivity, based on variable skin temperatures, is meaningfully comparable to the model-derived emissivity under the assumption of isothermal conditions. A brief sensitivity analysis or discussion of potential biases would improve the credibility of the comparison.
It would be valuable to assess how the derived emissivities vary with time of year and with seasonal temperature fluctuations.

We use a constant annual mean profile and compare the simulations with annual mean emissivities, so there is not mismatch due to seasonal propagation of the temperature at depth in our analysis. Averaging in time and using constant profile work if the dependency of Tb is linear to T(z) and if the thermal diffusion is linear (Surdyk 2002). This is usually the case in the Antarctic dry zone, considering the purpose of our study. This has been used in previous studies (Picard et al. 2009, Brucker al. 2010, and Picard et al. 2022).

More precisely, the diurnal and seasonal variations have been removed by averaging emissivities over a year, as mentioned in the text. To clarify how this averaging is performed, we reformulated the sentence as follows:
« To remove seasonal temperature variations, **the emissivities (calculated for each AMSR2 observation each day with the collocated ERA5 skin temperature) are averaged over the whole year of 2019. We only** averaged emissivities calculated with low cloud liquid water

content (less that $0.05~\mathrm{kg.m^{-2}}$, as indicated by the ERA5 reanalyses) to minimize cloud impact on the observed brightness temperatures. »

Line 109
"To remove seasonal temperature variations, AMSR2 data…"
Consider change "data" to "emissivity" to avoid confusion with brightness temperature.

Yes, AMSR2 data were misleading here. We confirm that we average each emissivity derived from the swath AMSR2 data, for all observations in 2019, with low cloud cover.

Line 130
It would benefit the reader if the authors could explicitly explain the physical reasons why Ku-band is more sensitive to scattering than C-band in this context.Line 146
Remove one "the"

We propose the following changes:
**"In general, scattering is stronger at higher frequencies for small scatterers, such as depth hoar in snow and air bubbles in the ice. This explains why QuikSCAT and OSCAT data (Ku-band) exhibit higher scattering than in ASCAT data (C-band). "**

Line 149
For clarity, it would be helpful to briefly explain what polydispersity is.

We have removed this sentence, as it is explained more extensively further in Section 3.2, with definitions.

Line 160
There appears to be a contradiction in this paragraph. The authors initially state that random structural variations are especially important in the top few meters, but then conclude that these do not significantly affect the microwave observations presented in Section 2. This claim seems unsupported. Rather than assuming these fluctuations are negligible, the authors could instead justify this by noting that the observations are averaged over a full year, which likely smooths out the influence of transient or small-scale random variability. Rephrasing to reflect this reasoning would improve logical consistency and credibility.

We propose to rephrased this paragraph to make it more explicit what "continuous" and "random" variations meant. It is now explicit with monotonic, centimeter-scale variations, and adding "vertical" where variations could have been understood as temporal variations. We have added references as well. The paragraph is:

**"**Our snowpack model assumes **monotonic variations in** grain size and density with depth**.** Field measurements have shown that models with continuously increasing density and grain size fit reasonably well the observed behaviors \citep{albert2004, courville2007, brucker2010, picard2014, leducleballeur2015, inoue2024}. These models do not account for the random **and centimeter-scale vertical variations \citep[e.g.][]{courville2007, leducleballeur2015, picard2014, inoue2024}, which arise due to the specific conditions during each snowfall event or storm, and are especially pronounced in the upper few**

**meters of the snowpack.** Rather than identifying a realistic snow profile corresponding to field measurements, as previously done by e.g. \citet{picard2022}, **we aim to capture the general behaviors in grain size and density over large regions.** We therefore assume that random and unresolved **centimeter-scale** fluctuations do not significantly affect the microwave observations presented in Section~\ref{sec:data} \citep{leducleballeur2015}. The assumption of continuous variations in grain size and density is less valid in H polarization, which is sensitive to **abrupt permittivity contrasts caused by density changes, and for higher frequencies, particularly 89 GHz, which probes only centimetric depths \citep{picard2012}. "**

Line 161-164
While the manuscript appropriately notes the limitations of assuming continuous vertical variations in grain size and density—especially in the context of H-polarized radiometry and high-frequency channels—it would significantly strengthen the analysis to include a quantitative estimate or sensitivity test illustrating how this assumption affects model performance. Even a simplified comparison could help assess whether this limitation introduces minor biases or fundamentally alters the interpretation of the signal.

While we acknowledge the need of such investigation in general, this proposition seems beyond the scope of our study for two reasons:

First, this would require significantly more material. To draw quantitative conclusions, realistic fluctuations are needed which can only be obtained from high resolution in situ measurements of density (and grain size), data that are scarce in the area of megadunes. Modeling H polarization has been addressed in other papers using high resolution profiles obtained at Concordia station, as cited in the paper. However, we acknowledge that the effect of smoothing has not been thoroughly investigated in the literature, except at L-band (Tan et al. 2015, Leduc-Leballeur et al. 2015).

The second reason relates to the objective of the study, which is to discuss the high backscatter / low emissivity issue on the icy moon and its Antarctic analog. Given current knowledge, medium fluctuations on the moons are of a secondary importance, until the general, smooth variations of the medium are better understood.

Related to the previous comment, we propose to add references to address this question.

Line 165
The manuscript describes a 10-layer snowpack model as a "continuous" profile. However, with discrete layers and potentially discontinuous density between adjacent layers, the profile is in fact piecewise continuous, and the SMRT model will simulate internal reflections at these interfaces. The authors should clarify whether such reflection effects are significant in their simulations, particularly for H-polarized channels.

The use of profiles with discrete layers could indeed have a small effect. This was tested and mentioned L182 in the initial manuscript: "Increasing the number of layers has no effect on the resulting emissivities and σ0, indicating that the chosen sampling is sufficient"

Line 167
The authors should explicitly state what material was used as the substrate in the SMRT simulations.

We meant the ice instead of the substrate. This is now corrected.

Line 169
The phrase "the same model has been used on Jupiter's icy moons by Brown et al. (2023)" is ambiguous. It would be helpful to specify whether "same model" refers to the use of the SMRT radiative transfer framework, the same physical configuration (e.g., snowpack structure, grain sizes, density profiles), or identical boundary conditions such as substrate properties. Clarifying this would help the reader understand the degree of comparability between the Earth-based and planetary modeling efforts.

We used the same equations for the ice dielectric properties. We propose to rewrite the sentence as follows:
"The dielectric property of the **ice** follows \citep[][p. 456--461]{matzler2006} **(the default in SMRT) which is valid for the frequency range of 0.01-3000 GHz and ice temperatures from 20 to 273.15 K: the same formulation has been used on Jupiter's icy moons by \citet{brown2023}"**

Line 183
For solid ice, is grain size still meaningful?

This information is not meaningful. It was added only because SMRT requires it, but it has no impact: setting the density of ice to 917 kg/m3 leaves no room for air bubbles (and grains size is irrelevant in this context). We removed this information.

Line 197-217
This section conflates multiple distinct scattering parameterizations available in SMRT, resulting in conceptual ambiguity. Specifically, it is unclear whether the authors are using the sticky hard sphere model—which introduces inter-particle interactions via a "stickiness" parameter—or a correlation length-based model, where scattering is governed by permittivity fluctuations described by quantities such as the Porod length and correlation length. These two approaches are physically distinct and are implemented independently in SMRT. If the authors are using a correlation-length-based scattering model, then references to "stickiness" are misleading and should be removed or rephrased appropriately.
Additionally, the paragraph references the method of Picard et al. (2022a, 2022b), but the implementation remains unclear. For instance, the sentence: "To use the correct value as correlation length, we use the microwave grain size, defined by Picard et al. (2022a)..." raises several questions. Is the microwave grain size being used directly as the correlation length lc? If not, what is the precise relationship between lc and lMW? Furthermore, the value K=0.62 from Picard et al. is described as "best for Antarctic snow," but was derived in which region of Antarctica? Can the authors justify applying it to the current study site?
To improve clarity, the authors should explicitly (1) state which scattering model in SMRT was selected (2) define how input parameters like optical grain size, microwave grain size, and correlation length are defined, derived and linked, and (3) avoid blending terminology

from physically incompatible models. Without this clarity, it is difficult to evaluate the validity or physical interpretability of the simulation results.

Regarding the stickiness, this term was used to explain how previous studies applied factors to adjust the grain size. Nevertheless, we agree that it could be removed.

Regarding the correlation length, "lc" was mistakenly used instead of "lp". The modified text is clearer and only mentions optical radius and microwave grain size.

Regarding the value K=0.62, it was found to work well at some sites in Antarctica and Canada for faceted crystals and fine grains (Picard et al. 2022). In the megadune region, depth hoar (which has a higher value of polydispersity) is likely more present, though not prevalent, as it is usually in the bottom layer in the Arctic. Our main results are largely insensitive to this value because of the fitting procedure, as stated in the text. This choice of K only affects the optical radius values, scaling it by a factor K_true / 0.62. Considering the other simplifications and the aim of the study, the impact of K is minor.

'Layer **optical radius:** Previous work has found that, for the grain sizes used in microwave radiometry simulations to be comparable to the optical **radius** measured in the field, a corrective factor must be used (Brucker et al., 2011; Royer et al.,2017). Recently, Picard et al. (2022b) has identified this factor as the polydispersity K, an intrinsic property of the snow microstructure, which can be measured from e.g. micro-computed tomography (Coléou et al., 2001). **More precisely**, we use the microwave grain size, defined by Picard et al. (2022b) as:

$$l_{MW} = K \times l_P \qquad (2)$$

where $l_P$ is the Porod length, calculated from the layer density $\rho$, ice density $\rho_{ice} = 917$ kg/m$^3$ and optical grain radius $r_{opt}$ as follows (Picard et al., 2022a):

$$l_P = 4/3(1 - \rho/\rho_{ice}) \, r_{opt} \qquad (3)$$

**where $r_{opt}$ is the radius of spheres having the same surface area over volume ratio as the considered snow microstructure (Grenfell and Warren, 1999). It is considered as a measurable quantity (Painter et al., 2007; Gallet et al., 2009; Picard et al., 2022a).**

Line 237
The manuscript should clarify whether the results presented in Figure 5 correspond to best-fit simulations under the stated assumptions. If so, please specify the method used to identify the best-fit model: was the parameter space (e.g., grain size, density, temperature) systematically explored using an optimization or sampling approach? Or were the parameter values selected based on physical reasoning or prior field measurements?

The parameter values are not intended as best-fit estimates. Instead, they are reasonable values determined from a combination of expertise and iterative trial-and-error. Avoiding automated optimization with a well-defined cost function fits our goal to explore the observation spectral behavior. For this purpose, we only need a "good enough" set of

parameters. We note that other values tend to degrade the fit, meaning that our solution is close to an effective best fit. We propose to explicitly emphasize the empirical nature of the selection as follows:

"The result of the simulation for AMSR2 V-polarized emissivities is shown in Fig.~\ref{fig:cst_rd} for physically reasonable **parameter values obtained by empirical trial-and-error; alternative values generally produce poorer fits.**"

Line 246
The author should explain why a crusted surface is necessary to explain the 89GHz data.

We propose to remove this distracting sentence, as this question is addressed in detail later in the manuscript.

Line 247-254
The manuscript compares the grain sizes retrieved in this study with those reported by Brucker et al. (2010), but the basis for this comparison is unclear. Are the two studies analyzing the same geographic region? Besides grain size, are other parameters (e.g., snow density, temperature, layering assumptions) held constant? Given that the model used in this work (Picard, 2022) incorporates more advanced microstructure parameterizations than models available in 2010, a direct comparison of grain size may not be meaningful. A more physically grounded comparison would involve correlation lengths or scattering coefficients. Furthermore, it would be informative to discuss how well the Brucker et al. model fits the current observations — do those earlier results underperform compared to the current model, or do they provide complementary insights? Clarifying these points would strengthen the validity of the comparison and help readers interpret the model advancement more clearly.

Brucker provides maps of optical radius grain size at the surface and its vertical gradient allowing us to make comparison in the same area. Note that this is more a consistency check than a comparison. We agree that the comparison is difficult, partly because of the different metrics used to measure the grain size: this is why we just mention the obtained parameter values briefly, and highlights the main conclusion, which is the ability to predict flat spectra at 19 and 37 GHz when using a constant density, which we found not possible when considering a large set of frequencies in our study.

We propose to update the paragraph to indicate that the comparison is in the same area. The differences in parameter values can be attributed to variations in modeling assumptions (Brucker used non sticky spheres, while we use optical radius).
"\citet{brucker2010} modeled the Antarctic snowpack at 19.3 and 37 GHz with increasing **optical radius** but constant density and found a reasonable match to the data (i.e., a roughly flat emissivity spectrum) **everywhere in the dry zone. We are also able to reproduce the data in the megadune area with the same grain size profile equation at the same frequency range,** although with different values for the parameters $r_{top}$ (**we find $r_{top}\approx 0.2$ mm; they find $0.45<r_{top}<0.65$ mm)** and $Q_2$ (**we find $Q_2\approx 0.06~\mathrm{mm^2.m^{-1}}$; they find $0.40<Q_2<0.82~\mathrm{mm^2.m^{-1}}$).** This discrepancy is due to modeling

differences between our simulations and those of \citet{brucker2010}. They used the DMRT-ML model composed of **non-sticky** spheres, whereas we use the SymSCE model while accounting for polydispersity (as defined by \citet{picard2022}). In any case, we show that while assuming constant density can be reasonable at 19.3 to 37 GHz**, our results demonstrate that it is inconsistent with observations over a wider spectrum range."**

Line 259
The phrase "several different datasets" is ambiguous. It would be helpful if the authors clarified what distinguishes these datasets — are they from different satellite instruments (e.g., active vs. passive), cover different frequency channels, represent distinct geographic regions, or span different time periods? Explicitly defining what is meant by "different" in this context would be helpful.
The author should also explain briefly how this range of parameter values are determined. What retrieval process is used?

By different datasets, we meant all satellite observations described in the Materials section. A more explicit sentence is:
**"Our goal is not to fit the model to the data, but instead to simulate the correct range of brightness temperatures and backscatter across all considered frequencies and polarizations, with the simplest possible snowpack model."**

We are not performing a 'retrieval' per se. The goal of this Section is to determine if it is possible to find a parameter set that simultaneously provides "the correct range of brightness temperatures and backscatter", for each grain profile shape (1$^{st}$, 2$^{nd}$ or 3$^{rd}$ order). The exact values of these parameters are not crucial for our objective. Our aim is to explore whether simulations with a simple snowpack can reproduce all considered frequencies and modes (active/passive). This is essential for applying the same approach to the icy moons where parameters are much less constrained and observations are sparse. While the specific parameter values are not very significant, it is important to ensure they fall in physically reasonable ranges. The main conclusion of this experiment is stated at the end of the section, in line with our objective: "We find that all considered microwave active and passive observations, except H-polarized emissivities, can be **simultaneously simulated by the SMRT with reasonable parameter values**."

Line 279
Q3 model
R^3(15m) = (0.1[mm])^3 + (50*1e-9 [mm3/m]) * 15[m] ◊ R(15m) ~ 0.1mm
Q2 model
R^2(15m) = (0.3[mm])^2 + 6e4*1e-6 [mm2/m]) * 15[m] ◊ R(15m) ~ 1 mm
"The grain radii we find in the subsurface are very large, with values around 1 mm at 15 m depths."
Please specify this is for the Q2 model and also note the corresponding value for the Q3 model.

There was an error in the legend of Figure 7 for the Q3 model. With the value 0.05 mm$^3$/m, we find 0.9 mm. The correct updated sentence is: **"The optical radii we find in the**

**subsurface are very large, with values around 1 mm at 15 m depths for $Q_2$, and 0.9 mm for $Q_3$.”**

Line 274
The statement that "ASCAT data, which is at lower frequency than 10–89 GHz AMSR2 and therefore probes deeper…" oversimplifies the depth-sensitivity comparison between active and passive sensors. Although ASCAT operates at lower frequencies, it is an active instrument and thus involves a two-way signal path, which significantly alters penetration behavior. Please consider rephrase this argument.
"can never be reproduced for the same configuration as AMSR2 data"
But in Fig6 panel C, the region between the two yellow lines does overlap with the observation range, why does the author make this statement?

We agree with the suggested reformulation. The initial sentence was overly simplified.

The statement concerned the $Q_1$ option only. The next sentence can be reformulated:
**"In contrast, the cubic increase ($Q_3$) and square increase ($Q_2$) can simulate** all data except H-polarized emissivities**, though the square option ($Q_2$) also has difficulties with the 89 GHz simulations (see Fig.~\ref{fig:res-best}).”**

Line 288
The manuscript suggests that the H-pol radiometric signal (AMSR2) cannot be reproduced due to polarization effects related to surface or subsurface layering. However, it is unclear why similar polarization effects would not also affect the active radar backscatter at HH polarization. Since both H-pol radiometry and HH-pol radar are sensitive to horizontal interfaces and anisotropy, some discussion of why the active signal can still be matched — while the passive signal cannot — would strengthen the interpretation.

We agree that the horizontally polarized waves, whether measured by passive or active instruments, should be similarly sensitive to the layering. Unfortunately, we do not have an explanation.  We propose to add a sentence to acknowledge the issue:
**"Radar observations in HH polarization should be equally affected, yet the simulations fall in the observed range in Fig.~\ref{fig:res-best}b. We do not have an explanation for this different behavior.”**

Line 298
"Indeed, it does not account for the observed ice crust in the wind-glazed regions, the variations of temperature with depth and season, and the random variations of density and grain size at these depths."
The explanation that the spectral slope cannot be reproduced due to "variations of temperature with depth and season" seems questionable, as the observations have already been averaged over an annual cycle. This averaging should significantly reduce the impact of seasonal thermal variability. Additionally, if random fluctuations in grain size and density were responsible, it is unclear why the slope would appear consistently across the entire region of interest — random features should not produce a coherent spatial signature. Moreover, in Section 4.3, the inclusion of an ice crust still does not recover the slope, which

suggests that other structural or radiative mechanisms may be at play. A more detailed investigation or alternative hypothesis may be warranted to explain this persistent model–data mismatch.

We agree and propose to remove the "temperature seasonal variations".
The hypothesis was that since the grain size in the upper centimeters may be seasonal, it is correlated with the temperature variations (e.g. smaller grain in winter, bigger grain in summer) which may not be averaged out over a year. This correlation does not affect the lower frequencies that are penetrating deeper. We propose to make more explicit the issue with the "random" variations.
The corrected sentence reads: **"**Indeed, it does not account for the observed ice crust in the wind-glazed regions and the **stochastic** variations of density and **optical radius at these depths that may have not been averaged using a single year of observations \citep{stefanini2024}. "**

Line 380
Please define NRCS.

"Normalized Radar Cross Section (NRCS)". The abbreviation has been avoided, and the full name is used instead.

Line 398
The application of a factor of 2 to the total SMRT backscatter to account for the Coherent Backscatter Opposition Effect (CBOE) appears problematic. The CBOE only enhances the multiple scattering component — not the total signal, which includes both single and multiple scattering. Since the SMRT output includes both contributions, applying a multiplicative factor to the full signal likely overestimates the CBOE enhancement. A more accurate approach would be to isolate the multiple scattering term (if possible within SMRT) and apply the enhancement selectively. I recommend revisiting this correction or clarifying its justification with reference to the physical assumptions and model limitations.

It is not our intention to "account for the Coherent Backscatter Opposition Effect", because of the difficulties evoked by the reviewer and in our manuscript in the conclusion. Instead, we intent to test the hypothesis already invoked in the literature whether CBOE could explain the Saturn passive-active observations or not, as described L396.

The answer to our test is "yes, it could" for some Saturn moons, and "yes it could improve but is insufficient" for Enceladus (L400). Our text makes it explicit twice that the factor of 2 is the theoretical maximum and we carefully crafted this paragraph to match our intention, and avoid overselling this result. The sentence L400 using "consistent" and "insufficient" is moderate.

Nevertheless, we would like to note here (but not in the main text) that the extremely high values of backscatter are indicative of extremely high single scattering albedo. In this case, cyclical multiple scattering mechanisms subject to CBOE are likely to represent a large proportion of the mechanisms, and the amplifying factor could approach the value of 2.

These results motivate further work to investigate this hypothesis using proper electromagnetic calculations.

Figure.8
The observed correlation between ASL and emissivity is clearly demonstrated, but the scatter in the relationship suggests that other physical factors are also influencing the outcome. It would be helpful if the authors briefly discussed the main contributors to this spread. Identifying which parameters have the strongest and weakest influence in this context would provide valuable insight, and help readers understand the limitations and sensitivities of the model.
As noted earlier, the SMRT simulated model over-estimates the emissivity in H pol. How is this effect going to affect your simulated correlation between ASL and e?
Please double check the units for Q1, Q2 and Q3 both in figures and text. There are quite a few places that the units are not correct.

Our ambitious in this late part of the discussion is only to address the "Implications for Saturn's icy moons" of our results. Adding new results, with a sensitivity analysis, would require to reconsider the organization of the paper (adding method, moving the analysis to the results) and its goal, for a benefit that is probably not great. In fact, the medium considered here, with only two layers, is over-simplified. It is simpler than that used for the Antarctic simulations. This limitation is adequate to highlight the broad relationship between the emissivity and the NRCS and test the hypothesis of the potential role of the CBOE, but to our opinion, it is insufficient to realistically explore the variability in this relationship.

Regarding the over-estimation of emissivity at H polarization, it is difficult to assess the impact for Figure 8, considering that 1) it is unlikely that layering on ice moons is as pronounced as in Antarctica, where every snowfall occurs in different conditions, so the emissivity may not be overestimated in the extraterrestrial context, 2) the figure represents a scatter plot with the radar in HH polarization as well, which is equally affected by layering.

Line 391
Based on the comparison presented in Fig6, the model using a Q3 grain size increase appears to fit the observations more closely than the Q2 model, with model parameters rtop=0.1mm, Q3=50um3/m. However, the text states that Q2 represents the best-fit model and with rtop=0.3mm, Q2=1e5um2/m (unit is wrong in the text, and in Fig6 Q2=6e4um2/m). This discrepancy should be addressed explicitly. If Q2 is preferred for reasons beyond data matching, this justification should be clearly stated. Otherwise, the statement about the best-fit model should be revised to align with the presented results.

This is indeed an error, when reporting the parameters. In a former iteration of the simulations, we used the Q2 model, but then moved to the Q3 model which provides the best fit. The results in Figure 8 are with the Q3 model, but the information had not been updated in the text.

The text is now updated and correct.

Line 412

The statement that the signal is driven by "incoherent scattering on large subsurface grains" is potentially misleading. In the Q2 model, the grain size ranges from 1 mm at 15 m depth to 3 mm at 100 m. These sizes are relatively small compared to the microwave wavelengths under consideration (1–6 cm), and may not qualify as "large" scatterers in scattering theory. Moreover, coherent scattering effects are generally most pronounced when scatterer sizes are comparable to the wavelength. The current phrasing is misleading, and I suggest reorganizing this argument to clarify the scale-dependence and the physical regime being invoked.

"**large**" was not necessary here, we propose to remove it. The message is that the signal is dominated by volume/grain scattering, and the way this scattering mechanism changes with depth.

Line 413
The statement that "an ice crust over the snowpack is necessary to reproduce H-polarized emissivities" appears too strong based on the evidence presented. While the inclusion of an ice crust lowers the modeled TB at H pol, it doesn't really improve agreement with observations. The paper does not demonstrate that this is the only viable configuration capable of doing so. Alternative explanations are not ruled out. I suggest rephrasing to indicate that the crust is a plausible or effective solution, rather than a proven requirement.

Our demonstration on this matter is indeed limited to testing whether adding a crust layer works or not. We propose to change the text: "Meanwhile, to reproduce H-polarized emissivities, **we found that adding an ice crust over the snowpack,** as observed in the wind-glazed regions, **is a viable configuration**". The following sentences clarify that this addition is far from sufficient.

---

## Author Comment (AC2)

**Response to Reviewer 2**

This work provides an active and passive simulation study for the East Antarctica using SMRT model. Simulations are done for a wide range of frequency channels from 5.2 to 89GHz. The authors want to draw an analogy between the ice moons and this particular region of Antarctica and looks like the authors want to claim that this region would be a good example for the study of icy moons.

From my personal perspective, some major points need to added to the paper and some concerns need to be resolved before the paper can be published.

We thank the reviewer for a thorough reading of our manuscript, and for constructive suggestions.

Here are some general comments for the paper:

1. In the abstract, the authors need to provide some conclusions that they obtained from this study and also need to provide the "up-shot" (how would this study contribute to a "larger picture" and would help answer a problem).

The larger picture is provided in the abstract which states that (i) the model well reproduces Antarctica data especially showing the need for an upper thin ice layer (ii) it helps explain the passive and active microwave observations of icy moons, but more work is to be done to interpret the very large backscattering signatures.

We added a sentence to the abstract to make it clearer:

"**More work is still to be done to fully reproduce the microwave signatures of icy surfaces in the solar system.**"

2. If the goal of the paper is to show that the Region of Interest (ROI) in the East Antarctica is a good analogy for icy moons, the active and passive data signature, and the measurement set-up for the icy moons needs to be presented and the features of the icy moons and ROI needs to be discussed. In such a way, the analogy could be drawn. Currently, the discussions are not sufficient.

One of the goals of the paper is to test whether the Antarctica ice sheet could serve as a good analog for the surfaces of Saturn's icy moons based on their measured active and passive microwave signatures. This question is valid because these surfaces share a common composition (dominated by water ice) and a potential common structure as the surfaces of icy moons are thought to be covered by a snow-like material originating from the E-ring (itself fed by Enceladus' geysers). This is explained in details in section 2.

Regarding the measurement set up for the icy moons, it is presented in length in Le Gall et al. (2023) as referenced in the Introduction section of the paper. It is out of the scope of this paper to repeat the (very long) description of the acquisition and reduction of this dataset.

To make it clearer on the origin of the icy moon data we added a sentence in the introduction:

"**The microwave data obtained on icy moons has been detailed in Le Gall et al. (2023)**"

3. Since the paper is majorly doing simulation to match up the observations, if parameters from icy moons can reproduce the measurements over ROI, this can also imply an analogy.

We are not sure to understand the reviewer's question/suggestion. One of the main goals of the paper was to test the Antarctica ice sheet as a potential analog for icy moons, not the other way around. The microwave observations of the Antarctica ice sheet are well reproduced by the model without having to invoke other parameters.

Detail comments are the following:

1. Resolution. As indicated by the sensor parameters, the scatterometers and radiometers are having different resolutions (ASCAT, Qscat 25km, AMSR2 based on frequency). In this work, the authors project the different data sets into uniform 12.5km grids. In such a way, the near by data pixels would be highly correlated and would not provide extra information for pixels within the resolution of a given data set. Such a interpolation would ignore the heterogeneity within a large resolution and may mistakenly use the coarse, larger area averaged measurement to represent the measurement for a smaller area. I believe a better way is to aggregate the high resolution data into low resolution such that different data sets can have the same averaging effect over the measured area. Can the authors provide some discussion on this?

The objective of this study is to provide a physical interpretation of multi-frequency passive and active microwave observations using reasonable geophysical parameters. While we acknowledge that observations from nearby pixels are likely spatially correlated for each observation type, this study does not attempt to explicitly exploit the spatial structure of the data.
When merging datasets with differing spatial resolutions, trade-offs are inevitable. As the reviewer suggested, one approach is to average the higher-resolution observations to match the lower-resolution dataset, specifically, the spatial resolution of the 6 GHz passive microwave channels. In this study, we chose to grid all observations to a common resolution of 12.5 km, which is close to the resolution of the 36 GHz AMSR2 channel. The goal is not to achieve a perfect match to each individual observation, but rather to ensure a consistent and physically reasonable interpretation across frequencies and observation modes. **The spatial resolutions of the different observations have been added in Table 1.**
Using a coarser grid (e.g., matching the 6 GHz resolution) would not alter the conclusions of the study. It would primarily smooth out some of the stronger scattering signatures observed at higher frequencies, at 36 and 89 GHz.

2. If my memory serves me correctly, L3 data from AMSR is already grided. That data set might be better? Only a suggestion.

In this study we use the L1R from AMSR2, at each native spatial resolution and at swath level as specified in section 2.1 in the paper. We do not make use of the L3 data from AMSR.

3.  The way of data averaging is not clear to me. How is the measured data averaged to a data point in each frequency?

This has been clarified in the text: **'For each observation type, the swath data are projected over a 12.5 km grid using the EASE-grid 2.0 Southern hemisphere grid projection (Brodzik et al., 2012, 2014). All pixels falling within a given grid point are averaged over a full year of data, for each instrument and observation conditions (frequency, polarization, incidence angle, and mode).'**

4.  In matching the data, active part looks fine to me, but the passive part doesn't look satisfactory. The observables from radiometers are brightness temperatures, emissivity values are derived values. Radiometers are very accurate, usually the errors are within 3K, assuming a physical temperature of 270K, this corresponds to an error in emissivity around 0.011. I would suggest the authors show the comparison in terms of brightness temperature. In such a way, the forward simulation would show a difference of 10K or more. Match up can be improved.

For passive microwave observations, we agree that the analysis could have been conducted directly using Tbs. However, emissivities were used in this study for two main reasons: 1) most studies on icy moons present results in terms of emissivity rather than Tbs, and 2) our team has extensive experience working with emissivity over snow- and ice-covered surfaces on Earth.
The relationship between emissivity and brightness temperature has been clarified in the manuscript, in section 2.1.
**'Note that at a frequency where the atmosphere is transparent, the radiative transfer equation reduces to $T_B = e \times T$, and a difference of 0.01 in emissivity $e$ with a snow / ice temperature of $T$ = 270 K results in a change of 3 K in $T_B$.'**

It is important to note that achieving agreement within 10 K across all passive microwave channels from 6 to 89 GHz and for both polarizations is already highly challenging. Even at a single frequency and polarization, significant discrepancies are common. For example, Burgard et al. (The Cryosphere, 2020) reported differences exceeding 10 K at 6 GHz V polarization over sea ice, despite multiple model adjustments. See below.

[Figure]

**Figure 4.** Observed brightness temperatures by AMSR-E (top row). Differences between brightness temperatures simulated with ARC3O from MPI-ESM output assimilated with SICCI2 (second row), Bootstrap (third row), and NASA Team (bottom row) sea ice concentration and observed brightness temperatures. The columns stand for the three cold seasons: JFM, AMJ, and OND. Summer (JAS) is discussed in Sect. 4.4.

At ECMWF, Hirahara et al. (Remote Sensing, 2020) performed simulations of Tbs over continental snow and compared them to AMSR2 observations from 6 to 89 GHz. The discrepancies observed were substantial, particularly for horizontal polarization, and increased significantly with frequency, both in terms of bias and standard deviation (see the red curves below).

These findings highlight the considerable challenge of achieving good agreement between simulated and observed passive microwave signals under frozen surface conditions, especially across a broad range of frequencies and polarizations. In this context, the level of agreement achieved in our study can be considered very acceptable.

[Figure]

**Figure 13.** Kernel density estimation of AMSR2 first guess departure (O−B) in K, for January 2019, for IFS:ATLAS (blue) and IFS:CMEM (red), at V (solid lines) and H (dotted lines), for snow-covered (excluding glacier and ice shelves) area without (left) and with (right) high vegetation, for the six AMSR2 frequencies from 6.9 GHz (top) to 89 GHz (bottom).

---

## Author Response (AR3)

**Reviewer 1:**

I do not agree with the authors' statement: "We agree with the reviewer that the factor 2 is an upper limit, and it has been made clear in the manuscript."
The factor of 2 can only be applied to the multiple-scattering component, not to the total backscattering value. Therefore, the yellow circles shown in Figure 8, which the authors describe as the "maximum theoretical value," are not correct—they overestimate the theoretical maximum.
I am not requesting that the authors explicitly model the CBOE effect. However, to make the results correct, I can only recommend acceptance if the authors choose one of the following options:

1. Estimate the single-scattering component theoretically, compute the multiple-scattering contribution, and apply the factor of 2 only to the multiple-scattering portion.
2. Explicitly acknowledge that the yellow circles in Figure 8 overestimate the theoretical maximum values, and explain why. (However, in this case, I question the meaning and usefulness of these yellow circles.)
3. Remove the yellow circles from Figure 8 and delete the related text.

We acknowledge that the yellow circles in Figure 8 overestimate the theoretical maximum values. The legend in the figure has been modified, and the corresponding text has been made clearer.

Figure 8. The results of SMRT simulations for a two-layer icy satellite model are shown, along with observations on Jupiter's and Saturn's icy satellites (from Le Gall et al., 2023) and simulated disk-integrated observations corresponding to the Antarctica megadunes region. Since the SMRT does not simulate the coherent backscatter opposition effect, $A_{SL}^{disk}$ is multiplied by 2 *to provide the upper theoretical limit of the simulations with the maximum possible COEB effect*. Even so, the model is unable to reproduce most Cassini observations of Saturn's icy moons.

'*In theory, the CBOE can at most multiply the radar returns ($\sigma^0$ and therefore also $A^{disk}$) by a factor of 2. To investigate the upper limit of SMRT simulations if they were including CBOE, we plot the simulations multiplied by 2, and find that these are consistent with some Saturn moon observations, but still insufficient for Enceladus, the radar-brightest object in the solar system. This exercise reinforces the hypothesize that CBOE is necessary to explain moon backscatter, although it may not fully explain the behavior of the brightest moons. Integrating CBOE into SMRT is a meaningful avenue for future improvement*.'

**Reviewer 2:**

After reviewing the paper, I believe the authors have addressed most of my previous questions. There is only one comment I want to make and just FYI. The paper discussed the difference in V and H observations. There are several papers discussing this phenomenon. Some tends to interpret this effect by horizontal variations using 3D random media. Some people use internal rough surfaces to explain the V and H difference.

We acknowledge that the topic of polarization differences has been explored in several previous works, resulting in varying interpretations of the observed effects. We thank the reviewer for this additional comment.